# DECLARATIVE CHARACTERIZATIONS OF DIRECT PREFERENCE ALIGNMENT ALGORITHMS

## ABSTRACT

Recent direct preference alignment algorithms (DPA), such as `DPO`, have shown great promise in aligning large language models to human preferences. While this has motivated the development of many new variants of the original DPO loss, understanding the differences between these recent proposals, as well as developing new DPA loss functions, remains difficult given the lack of a technical and conceptual framework for reasoning about the underlying semantics of these algorithms. In this paper, we attempt to remedy this by formalizing DPA losses in terms of discrete reasoning problems. Specifically, we ask: *Given an existing DPA loss, can we systematically derive a symbolic expression that characterizes its semantics? How do the semantics of two losses relate to each other?* We propose a novel formalism for characterizing preference losses for single model and reference model based approaches, and identify symbolic forms for a number of commonly used DPA variants. Further, we show how this formal view of preference learning sheds new light on both the size and structure of the DPA loss landscape, making it possible to not only rigorously characterize the relationships between recent loss proposals but also to systematically explore the landscape and derive new loss functions from first principles. We hope our framework and findings will help provide useful guidance to those working on human AI alignment.

## 1 INTRODUCTION

Symbolic logic has long served as the de-facto language for expressing complex knowledge throughout computer science (Halpern et al., 2001), including in AI (McCarthy et al., 1960; Nilsson, 1991), owing to its clean semantics. Symbolic approaches to reasoning that are driven by declarative knowledge, in sharp contrast to purely machine learning-based approaches, have the advantage of allowing us to reason transparently about the behavior and correctness of the resulting systems. In this paper we focus on the broad question: *Can the declarative approach be used to better understand and formally specify algorithms for large language models (LLMs)?*

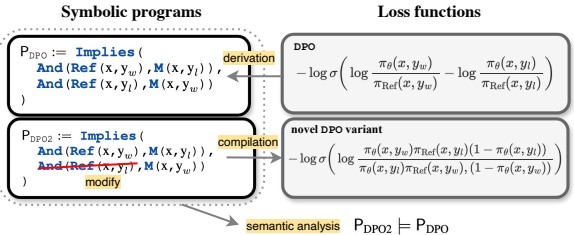

Figure 1: Can we uncover the hidden logic of `DPO`? Here we show the distillation of the `DPO` loss into a symbolic expression that expresses its high-level model behavior, along with a modified version of that program that we can compile into a novel `DPO` loss.

We specifically investigate direct preference learning algorithms, such as direct preference optimization (DPO) (Rafailov et al., 2024), for pairwise preference learning, which are currently at the forefront of research on LLM alignment and learning from human preferences (Ouyang et al., 2022; Wang et al., 2023). While there has been much recent work on algorithmic variations of `DPO` (Azar et al., 2023; Hong et al., 2024; Meng et al., 2024, *inter alia*) that modify or add new terms to the original loss, understanding the differences between these new proposals, as well as coming up with new variants, remains a formidable challenge due to the lack of a conceptual and technical framework for reasoning about their underlying semantics.

Our study attempts to remedy this problem by formalizing the corresponding loss functions in terms of logic. Such a formalization is based on trying to answer the question: *Given an existing loss function, such as DPO (see Figure 1), can we derive a symbolic expression that captures the core semantics of that loss function (i.e., one that we can then systematically compile back into the exact loss)?* In treating loss functions as discrete reasoning problems, ones that abstract away from lower-level optimization details and tell us about high-level model behavior, it becomes possible to study them using conventional semantic notions from logic (e.g., *entailment*), relate it semantically to other programs, or even modify its underlying logical semantics to derive entirely new algorithms.

To do formalization, we devise a novel probabilistic logic based on a generalization of the notion of *semantic loss* (SL) (Xu et al., 2018) coupled with a provably correct mechanical procedure for translating existing DPA losses into programs in our logic. As in SL, losses are produced from symbolic programs by counting the weighted propositional models of those programs, reducing the problem to one of probabilistic inference (Chavira & Darwiche, 2008). In contrast to the kinds of symbolic programs commonly used with SL, however, empirically successful DPA losses impose systematic conditional constraints on the types of models that should be counted, which shape the structure of the underlying probability distribution. We express these constraints through a new primitive in our logic called a **preference structure** that also addresses various technical and conceptual issues involved with modeling pairwise preference symbolically. It is through such constraints that certain semantic relationships between existing losses can be easily observed and new losses can be derived.

Our formal view of preference learning sheds much light on the size and structure of the **DPA loss landscape**. Under modest assumptions motivated by the structure of existing DPA losses and our new logic, we see that the number of definable DPA losses is doubly exponential over the number ($n$) of unique predictions (i.e., forward model calls) made in a loss function, or $4^{2^n}$. This results in, for example, close to 4.3 billion unique variations of the original DPO loss, which leaves much room for exploration. While big, we show how this space is structured in interesting ways based on formal connections between relationships that hold in the semantic space among formalized DPA losses (e.g., logical entailment, equivalence) and their monotonicity properties in the loss space.

These formal results also provide practical insights into how to effectively search for new DPA losses. For example, one can start with empirically successful loss functions, use the formalization to understand their semantics, then modify their semantics to arrive at novel variants that are either more constrained or less, then experiment accordingly.

## 2 RELATED WORK

**Language model alignment** While traditional approaches to language model alignment have employed reinforcement learning (Ziegler et al., 2019; Christiano et al., 2017), we focus on DPA approaches such as DPO (Rafailov et al., 2024) and SliC (Zhao et al., 2023) that use closed-form loss functions to tune models directly to offline preferences.

We touch on two recent areas in this space: formal characterizations of DPA losses (Azar et al., 2023; Tang et al., 2024; Hu et al., 2024) and work on devising algorithmically enhanced variants of DPO (Amini et al., 2024; Hong et al., 2024; Meng et al., 2024; Pal et al., 2024; Xu et al., 2024; Ethayarajh et al., 2024; Park et al., 2024). In contrast to this work on formal characterization, which focuses on the optimization properties of DPA losses and particular parameterizations like Bradley-Terry, we attempt to formally characterize the semantic relationships between these variants of DPO in an optimization agnostic way to better understand the structure of the DPA loss landscape.

**Neuro-symbolic modeling** For formalization, we take inspiration from work on compiling symbolic formulas into novel loss functions (Li et al., 2019; Fischer et al., 2019; Marra et al., 2019; Asai & Hajishirzi, 2020, *inter alia*), which is used for incorporating background constraints into learning to improve training robustness and model consistency. In particular, we focus on approaches based on probabilistic logic (Manhaeve et al., 2018; Ahmed et al., 2022; 2023; van Krieken et al., 2024).

In contrast to this work, however, we focus on the inverse problem of **decompilation**, or deriving symbolic expressions from known and empirically successful loss functions to better understand their semantics (see Friedman et al. (2024) for a similar idea). Work in this area has mostly been limited to symbolically deriving standard loss function such as cross-entropy (Giannini et al., 2020; Li et al., 2019), whereas we look at deriving more complex algorithms for LLMs.

## 3 DIRECT PREFERENCE ALIGNMENT

In this section, we review the basics of offline preference alignment, which can be defined as the following problem: given data of the form: $D_{\mathrm{p}} = \left\{ (x^{(i)}, y_w^{(i)}, y_l^{(i)}) \right\}_{i=1}^{M}$ consisting of a model input $x$ and two possible generation outputs (often ones rated by humans), a preferred output $y_w$ (the *winner w*) and a dispreferred output $y_l$ (the *loser l*), the goal is to optimize a policy model (e.g., an LLM) $y \sim \pi_\theta(\cdot \mid x)$ to such preferences.

As mentioned at the outset, we focus on direct preference alignment (DPA) approaches that all take the form of some closed-form loss function $\ell$ that we can use to directly train our model on $D_{\mathrm{p}}$ to approximate the corresponding ground preference distribution $p^*(y_w \succ y_l \mid x)$ (where $y_w \succ y_l$ denotes that $y_w$ is preferred over $y_l$). Since our study focuses on the formal properties of DPA losses, it is important to understand their general structure, which will take the following form (Tang et al., 2024):

| | $f(\rho_\theta, \beta) =$ | $\rho_\theta$ (standard formulation) |
|---|---|---|
| DPO | $-\log \sigma(\beta \rho_\theta)$ | $\log \frac{\pi_\theta(y_w\mid x)}{\pi_{\mathrm{ref}}(y_w\mid x)} - \log \frac{\pi_\theta(y_l\mid x)}{\pi_{\mathrm{ref}}(y_l\mid x)}$ |
| IPO | $(\rho_\theta - \frac{1}{2\beta})^2$ | |
| SliC | $\max(0, \beta - \rho_\theta)$ | $\log \frac{\pi_\theta(y_w\mid x)}{\pi_\theta(y_l\mid x)}$ |
| RRHF | $\max(0, -\rho_\theta)$ | $\log \frac{\pi_\theta(y_w\mid x)^{\frac{1}{\mid y_w\mid}}}{\pi_\theta(y_l\mid x)^{\frac{1}{\mid y_l\mid}}}$ |

Table 1: Examples of some popular DPA loss functions with different choices of $f$ and $\rho_\theta$.

$$\ell_{\mathrm{DPA}}(\theta, D) := \mathbb{E}_{(x,y_w,y_l) \sim D_{\mathrm{p}}} \left[ f\big( \rho_\theta(x, y_w, y_l), \beta \big) \right] \tag{1}$$

consisting of some convex loss function $f : \mathbb{R} \times \mathbb{R}+ \to \mathbb{R}$, a model quantity $\rho_\theta(x, y_w, y_l)$ which we will abbreviate to $\rho_\theta$ and a parameter $\beta$.[1]

Table 1 lists four specific DPA losses: DPO (Rafailov et al., 2024), IPO (Azar et al., 2023), SliC (Zhao et al., 2022; 2023), and RRHF (Yuan et al., 2023). Here the logistic log loss (shown using the logistic function $\sigma(x) = \frac{1}{1+\exp(-x)}$), square loss, hindge loss, and perceptron loss are used for $f$, respectively. Loss functions such as SliC and RRHF are examples of single model approaches that define $\rho_\theta$ in terms of the **log ratio of the winner and loser** given prediction probabilities $\pi_\theta$ of the model being trained. As an important implementation detail, the prediction probabilities are sometimes computed using **length normalization** as shown for RRHF. For DPO and IPO, in contrast, the model quantity $\rho_\theta$ is the **log ratio difference** (of the winner and the loser) between the predictions of the model being trained and a frozen LLM called a reference model, $\pi_{\mathrm{ref}}$. These two approaches constitute a two model approach, where the role of the reference model is to avoid overfitting on the target preference data (controlled by the parameter $\beta$).

Single model approaches have the advantage of avoiding the overhead associated with having an additional reference model and can sometimes yield competitive performance when compared against two model approaches. In the absence of a reference model, these losses are usually regularized using an added cross-entropy term, which we exclude from our formal analysis.

**The structure of DPA variants.** Conceptually, preference losses involve making predictions about winners and losers across models and reasoning about the relationships between predictions. The main question we ask is: *If we view this process as a discrete reasoning problem, what is the nature of the reasoning that underlies these different losses and each $\rho_\theta$?* To do our analysis, we start by rewriting each loss function in a way that strips away optimization and implementation details (e.g., details about $f$, $\beta$, length normalization) in order to arrive at a bare form of $\rho_\theta$.

Accordingly, we will write $P_m(y \mid x)$ in place of $\pi_\theta(y \mid x)$ to denote the probability assigned by a model $m$ to an output $y$ in a way that is agnostic to whether length normalization is used. In Table 2, we show different variants of DPO that we consider and two common baselines, the cross-entropy loss $\ell_{\mathrm{CE}}$ and a variant that uses an unlikelihood (Welleck et al., 2019) term $\ell_{\mathrm{CEUnl}}$. Importantly, we later express each $\rho_\theta$ as a single log ratio $\rho_\theta^t / \rho_\theta^b$, which we refer to as the **core loss equation**.

To more easily see the relationships between these proposals, we rewrite each $\rho_\theta$ in terms of the log ratio function $s_m(y_1, y_2)$ defined in Table 2 (we use $\overline{y}$ to denote the negation of $y$, or $1 - P_m(y \mid x)$).

---

[1] Following Tang et al. (2024) and their GPO framework, we formulate DPA approaches as general binary classification problems and do not make any assumptions about the preference structure $p(y_w \succ y_l \mid x)$.

Here we see that all losses are derivable from the log ratio of winner and loser $s_\theta(y_w, y_l)$ used in `SliC` and `RRHF` either exactly, as in `CPO` (Xu et al., 2024), or with added terms. `DPO`, for example, is expressible as this ratio minus an additional log ratio term $s_{\text{ref}}(y_w, y_l)$ that contains information about the reference model. Many variations to `DPO` then involve making the following two modifications.

**Adding additional terms.** Approaches like $\ell_{\text{DPOP}}$ (Pal et al., 2024) (see also Amini et al. (2024); Park et al. (2024)) incorporate additional terms into `DPO` ($s_{\text{ref}2,\theta2}(y_w, y_w)$) that address particular failure cases. We use $\theta2$ and ref2 to refer to copies of our two models, which is a decision that we address later when discussing the structure of the equation class assumed for $\rho_\theta$ (Section 5.2) .

| Loss | $\rho_\theta := \log \frac{\rho_\theta^t}{\rho_\theta^b}$ | $s_{m_1(,m_2)}(y_1, y_2) := \log \frac{P_{m_1}(y_1|x)}{P_{m_2}(y_2|x)}$ | |
|---|---|---|---|
| **Baselines** $\rho_\theta$ | | | |
| $\ell_{\text{CE}}$ | $\log \frac{P_\theta(y_w|x)}{1-P_\theta(y_w|x)}$ | $\ell_{\text{CEUnl}}$ $\log \frac{P_\theta(y_w|x)(1-P_\theta(y_l|x))}{P_\theta(y_l|x)+(1-P_\theta(y_w|x)))}$ | |
| **Single model approaches (no reference)** $P_\theta$ | | | |
| $\ell_{\text{CPO}}$ | $\log \frac{P_\theta(y_w|x)}{P_\theta(y_l|x)}$ | $s_\theta(y_w, y_l)$ | |
| $\ell_{\text{ORPO}}$ | $\log \frac{P_\theta(y_w|x)(1-P_\theta(y_l|x))}{P_\theta(y_l|x)(1-P_\theta(y_w|x))}$ | $s_\theta(y_w, y_l)$ | $-s_\theta(\overline{y_w}, \overline{y_l})$ |
| $\ell_{\text{SimPO}}$ | $\log \frac{P_\theta(y_w|x)P_{\text{mref}}(y_l|x)}{P_{\text{mref}}(y_w|x)P_\theta(y_l|x)}$ | $s_\theta(y_w, y_l)$ | $-s_{\text{mref}}(y_w, y_l)$ |
| **with reference model** $P_{\text{ref}}$ | | | |
| $\ell_{\text{DPO}}$ | $\log \frac{P_\theta(y_w|x)P_{\text{ref}}(y_l|x)}{P_{\text{ref}}(y_w|x)P_\theta(y_l|x)}$ | $s_\theta(y_w, y_l)$ | $-s_{\text{ref}}(y_w, y_l)$ |
| $\ell_{\text{DPOP}}$ | $\log \frac{P_\theta(y_w|x)P_{\theta2}(y_w|x)P_{\text{ref}}(y_l|x)}{P_{\text{ref}}(y_w|x)P_{\text{ref}2}(y_w|x)P_\theta(y_l|x)}$ | $s_\theta(y_w, y_l)$ | $-s_{\text{ref}}(y_w, y_l)$ $-s_{\text{ref}2,\theta2}(y_w, y_w)$ |

Table 2: How are variants of `DPO` structured? Here we define some popular variants in terms of their **core loss equation** $\rho_\theta$ and the helper function $s_{m_1,m_2}(y_1, y_2)$ (last column) that rewrites each $\rho_\theta$ in a way that brings out general shared structural patterns and added terms compared with the log win/loss ratio $s_\theta(y_w, y_l)$.

**Changing the reference ratio. No reference** approaches, such as $\ell_{\text{ORPO}}$ (Hong et al., 2024) and $\ell_{\text{SimPO}}$ (Meng et al., 2024) instead reparameterize the reference ratio $s_{\text{ref}}(y_w, y_l)$ either in terms of some quantity from our policy model as in `ORPO` ($s_\theta(\overline{y_w}, \overline{y_l})$) or a heuristic penalty term $\gamma$ as in `SimPO`. For `SimPO` we rewrite $\gamma$ term in terms of the ratio $\gamma = s_{\text{mref}}(y_w, y_l)$ (where 'mref' refers to a *manual* reference model) to make it align to `DPO`. For example, given any $\gamma \geq 0$ and manual $P_{\text{mref}}(y_w \mid x)$, $\gamma = s_{\text{mref}}(y_w, y_l)$ can be satisfied by setting $P_{\text{mref}}(y_l \mid x) = P_{\text{mref}}(y_w \mid x)/\exp(\gamma)$.

While our techniques will cover both reference and no reference approaches, due to their simplicity and the ability to derive the former from the latter, we use no reference losses such as $\ell_{\text{CEUnl}}$, $\ell_{\text{CPO}}$, $\ell_{\text{ORPO}}$ and a novel loss $\ell_{\text{unCPO}}$ (defined later) as running examples throughout.

# 4 PREFERENCE MODELING AS A REASONING PROBLEM

To better understand the DPA loss space, we will formalize the preference losses and the model quantities $\rho_\theta$ introduced in the previous section in terms of symbolic reasoning problems. This will involve the following core ideas and assumptions.

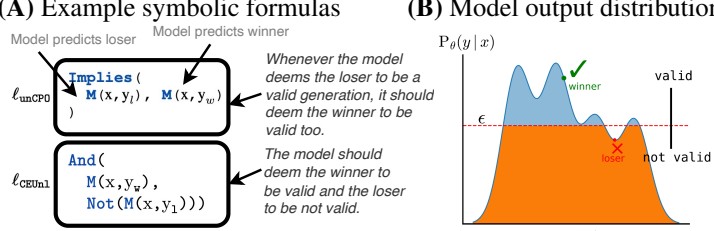

**(A)** Example symbolic formulas   **(B)** Model output distribution

**Model predictions are symbolic objects** The declarative approach will involve thinking of LLMs predictions as logical propositions. For example, when a model **M** generates an output $y_w$ for a prompt $x$, we will use the notation $\mathbf{M}(x, y_w)$ to express the proposition that $y_w$ *is a valid generation for* $x$. Importantly, we will further weight these propositions by assigning the probabilities given by our LLMs, i.e., $P_\theta(\mathbf{M}(x, y_w)) = P_\theta(y_w \mid x)$. We call these our **probabilistic predictions** $X_1, ..., X_n$, which will form the basis of symbolic formulas.

Figure 2: What do formal representations of loss functions tell us? We show (A) two symbolic formulas related to single model preference learning with their semantics in English. When grounded in model behavior, they tell us about the structure of the model's output probability distribution (B) and where predictions belong in that distribution (relative to some $\epsilon$). We will later show that these formulas correspond to the losses $\ell_{\text{unCPO}}$ (Figure 4) and the common baseline $\ell_{\text{CEUnl}}$ (Table 2).

**Relationships between predictions are expressed as symbolic formulas**   Relationships between model predictions will take the form of symbolic constraints expressed as formulas of propositional logic P defined by applying zero or more Boolean operators over probabilistic predictions. For example, in Figure 3 (A), the top formula, which we later show is fundamental to the semantics of many DPA approaches, uses the implication operator (**Implies**) to express the constraint that model **M** should never deem the loser $y_l$ to be a valid generation ($\mathbf{M}(x, y_l)$) without deeming the winner $y_w$ to also be valid ($\mathbf{M}(x, y_w)$). The bottom formula tells us instead that only the winner $y_w$ should be deemed valid using the conjunction and negation operators (**And**, **Not**).[2]

When grounded to model behavior via the proposition weights, such constraints tell us about the structure of a model's output probability distribution, as visualized in Figure 3 (B). Semantically, we assume that what constitutes a valid generation is any probabilistic prediction whose weight exceeds some threshold $\epsilon$ in that distribution, similar to $\epsilon$-truncated support in Hewitt et al. (2020). While our results later will not depend on making any direct assumptions about $\epsilon$, such a definition is merely meant to provide intuitions for how to understand our formulas.

**Loss functions are expressible as symbolic formulas**   We assume that all preference loss functions have an internal logic that can be expressed in the form described above. Our main goal is to uncover that internal logic, and to use semantic concepts, such as entailment (denoted as $\models$) or logical equivalence ($\equiv$) to meaningfully characterize the DPA loss space.

### 4.1 COMPILATION AND DECOMPILATION

**Compilation and semantic loss**   To compile a symbolic formula P into loss, we employ a probabilistic approach based on the semantics of a variant of weighted model counting (WMC) (Chavira & Darwiche, 2008; Fierens et al., 2015). This is based on computing a probability of a formula P:

$$p_\theta(\mathsf{P}) = \mathrm{WMC}(\mathsf{P}; \theta) := \sum_{\mathbf{w} \in \{0,1\}^n} \mathbb{1}\{\mathbf{w} \models \mathsf{P}\} \prod_{\mathbf{w} \models X_i} P_\theta(X_i) \cdot \prod_{\mathbf{w} \models \neg X_i} \big(1 - P_\theta(X_i)\big) \qquad (2)$$

or as a weighted sum over all the propositional models of that formula $\mathbf{w} \models \mathsf{P}$, or truth assignments (e.g., rows in the truth table in Figure 3 where P is satisfied ($\checkmark$)). Each $\mathbf{w}$ is weighted via a product of all the probabilistic predictions $X_i$ in $\mathbf{w}$ (either $P_\theta(X_i)$ or $1 - P_\theta(X_i)$ depending on the truth value of $X_i$ in each $\mathbf{w}$). A loss can then be obtained by taking the negative logarithm of this probability, which is known as the semantic loss first defined in Xu et al. (2018).

Formally, the semantic loss takes the form $\mathbb{E}_{d \sim D}[-\log p_\theta(\mathsf{P}_d)]$, where we use the notation $\mathsf{P}_d$ throughout to refer to the substitution of variables in our formulas P (e.g., $x, y_w, y_l$) with specific values from $d \sim D$. Since our approach will later involve computing the probability of P conditioned (optionally) on some **conditioning constraints** $\mathsf{P}_\mathbf{C}$ (i.e., an additional propositional formula), we consider the conditional form of the semantic loss and show its full objective below:

$$\min_\theta \mathbb{E}_{d \sim D} \left[ -\log p_\theta(\mathsf{P}_d \mid \mathsf{P}_{\mathbf{C}_d}) \right], \quad p_\theta(\mathsf{P} \mid \mathsf{P}_\mathbf{C}) = \frac{\mathrm{WMC}(\mathsf{P} \wedge \mathsf{P}_\mathbf{C}; \theta)}{\mathrm{WMC}(\mathsf{P} \wedge \mathsf{P}_\mathbf{C}; \theta) + \mathrm{WMC}(\neg\mathsf{P} \wedge \mathsf{P}_\mathbf{C}; \theta)} \qquad (3)$$

where the last part follows from the standard definition of conditional probability (with the denominator being an expanded form of $\mathrm{WMC}(\mathsf{P}_\mathbf{C}; \theta)$). We note that when $\mathsf{P}_\mathbf{C}$ is equal to $\top$ (or true), this form of the semantic loss is equivalent to the original version.

As an important technical point, we see below how having an explicit negation $\neg\mathsf{P}$ in the normalization allows us write the probability of P in the following way (without loss of generality, we exclude $\mathsf{P}_\mathbf{C}$ to improve readability and remove $\theta$ from WMC):

$$p_\theta(\mathsf{P}) = \frac{\exp\big(\log \mathrm{WMC}(\mathsf{P})\big)}{\exp\big(\log \mathrm{WMC}(\mathsf{P})\big) + \exp\big(\log \mathrm{WMC}(\neg\mathsf{P})\big)} = \sigma\Bigg( \underbrace{\log \frac{\mathrm{WMC}(\mathsf{P})}{\mathrm{WMC}(\neg\mathsf{P})}}_{\textbf{semantic loss ratio}} \Bigg) \qquad (4)$$

$$\text{with } \ell(\mathsf{P}, \theta, D) := \mathbb{E}_{d \sim D} \left[ -\log p_\theta(\mathsf{P}_d) \right] = \mathbb{E}_{d \sim D} \left[ -\log \sigma\Bigg( \log \frac{\mathrm{WMC}(\mathsf{P}_d)}{\mathrm{WMC}(\neg\mathsf{P}_d)} \Bigg) \right] \qquad (5)$$

---

[2]We will switch between using conventional logical notation (e.g., $\wedge, \vee, \neg, \rightarrow, \oplus$) and operator notation (e.g., **And**, **Or**, **Not**, **Implies**, **XOR**) depending on the context.

yielding a logistic log form of the semantic loss $\ell(\mathsf{P}, \theta, D)$ that aligns with the structure of the DPA losses in Section 3. As an analog to $\rho_\theta$, we call the inner part of $\sigma(\cdot)$ above the **semantic loss ratio**.

**Decompilation** The goal of decompilation is to derive for a loss function $\ell_x$ a symbolic expression P that characterizes the semantics of that loss. As we show later in Sec. 5.2, this will reduce to the problem of finding a program whose *semantic loss ratio* is equivalent to a loss's *core loss equation* $\rho_\theta$, based largely on the derivation above and its connection with DPA.

# 5 A LOGIC FOR PREFERENCE MODELING

In the standard semantic loss (SL), ML loss functions $\ell_x$ are expressible as a single propositional formulas P interpreted via probabilistic logic, with $\ell_x \sim -\log p_\theta(\mathsf{P})$. At first glance, this formulation is at odds with standard formulations of pairwise preference, such as the Bradley-Terry (BT) model (Bradley & Terry, 1952) typically assumed in RLHF, which involves modeling a preference distribution $p_\theta(y_w \succ y_l)$ between two (often disparate) quantities (e.g., given by the kinds of log ratios in Table 2). Indeed, logical accounts of pairwise preference such as Jeffrey (1965); Rescher (1967) assume a similar semantics where preference is defined not as a single propositional formula but as and inequality between model counts $\mu$ of two independent formulas $\mu(\mathsf{P}_w) > \mu(\mathsf{P}_l)$.

We observe none of the DPA losses in Table 2 and their log ratios can be expressed as a single propositional formula in standard SL using only their probabilistic prediction variables[3] While this can be remedied by creating a new version of SL that involves counting multiple formulas as in Rescher (1967), we instead define a relational structure and encoding called a **preference structure** that allows us to capture the semantics of losses in a modular fashion using a single propositional formula coupled with auxiliary constraints. Such a structure, which is based on a novel construction in propositional logic for encoding multiple formulas, will later make it easy to cleanly characterize different DPA losses and gives rise to a generalized form of SL (see Figure 3 for a high-level illustration).

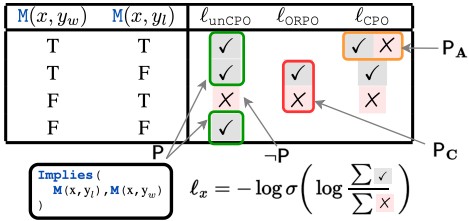

Figure 3: The Boolean semantics (top) of our version of semantic loss and preference structures: $\checkmark$ correspond to propositional models satisfying $\mathsf{P}$, $\overline{\mathsf{P}_f}$, $\times$ s to $\neg\mathsf{P}$ and $\overline{\neg\mathsf{P}_f}$, blank cells to conditioning constraints $\mathsf{P_C}$ and cells with multiple marks to $\mathsf{P_A}$. Losses (columns) are created by assigning/removing marks then counting these marks/rows via WMC and using the the bottom Eq. (following from Eq. 5).

**Preference structure** A preference structure is a tuple $\overline{\mathsf{P}} = (\mathsf{P}, \mathsf{P_C}, \mathsf{P_A})$ consisting of three propositional formulas: a **core semantic formula** $\mathsf{P}$ coupled with **conditioning constraints** $\mathsf{P_C}$ (as in Eq 3, which restrict the propositional models that can be counted) and **additive constraints** $\mathsf{P_A}$ that tell us what propositional models always need to be counted. As we will show, all the DPA losses in Table 2 are representable as preference structures, often ones where the same core formula $\mathsf{P}$ is shared (e.g., the formulas in Figure 3), yet that differ in the constraints they impose ($\mathsf{P_C}$ and $\mathsf{P_A}$).

Each preference structure will have a **formula form** $\overline{\mathsf{P}_f}$ and a **negated formula form** $\overline{\neg\mathsf{P}_f}$, which are defined by the following two propositional formulas (see running examples in Figure 3):

$$\overline{\mathsf{P}_f} := \Big(\mathsf{P} \vee \mathsf{P_A}\Big) \wedge \mathsf{P_C}, \quad \overline{\neg\mathsf{P}_f} := \Big(\neg\mathsf{P} \vee \mathsf{P_A}\Big) \wedge \mathsf{P_C}. \tag{6}$$

In the absence of the additive constraint $\mathsf{P_A}$, we note that these representations encode the conditional $\mathsf{P} \mid \mathsf{P_C}$, thus making the semantic loss of these formulas equivalent to the conditional semantic loss in Eq 3. Indeed, many DPA losses will be reducible to the conditional semantic loss, however, $\mathsf{P_A}$ and the ability to add default model counts to $\mathsf{P}$ and $\neg\mathsf{P}$ will be needed to derive some DPA losses symbolically and account for peculiar properties of their normalization.

---

[3]To see this for the ratio $s_\theta(y_w, y_l)$ from Table 2, which has two probabilistic prediction variables $y_w$ and $y_l$, one can enumerate all 16 unique Boolean functions over variables $y_w$ and $y_l$ to see that none yield a semantic formula whose WMC is equal to $\sigma(s_\theta(y_w, y_l))$. Through further analysis, one can also see that it is not possible to derive $\sigma(s_\theta(y_w, y_l))$ using conditional WMC either. The same argument can be applied to other losses.

Below we show that any two propositional formulas can be expressed as a preference structure based on a particular construction, called the **implication form**, that we use later for decompilation.

**Proposition 1.** *Given any two propositional formulas* $P_1$ *and* $P_2$*, there exists a preference structure* $\overline{P}$ *such that* $P_1 \equiv \overline{P_f}$ *and* $P_2 \equiv \overline{\neg P_f}$*.*

*Proof.* We provide a specific construction we call the **implication form** of $P_1$ and $P_2$. This is based on the following logical equivalences (the correctness of which can be checked manually):

$$P_1 \equiv \left( \underbrace{(P_2 \to P_1)}_{P} \vee \underbrace{(P_1 \wedge P_2)}_{P_{\mathbf{A}}} \right) \wedge \underbrace{(P_1 \vee P_2)}_{P_{\mathbf{C}}}, P_2 \equiv \left( \underbrace{\neg(P_2 \to P_1)}_{\neg P} \vee \underbrace{(P_1 \wedge P_2)}_{P_{\mathbf{A}}} \right) \wedge \underbrace{(P_1 \vee P_2)}_{P_{\mathbf{C}}}$$

As noted above, this construction corresponds exactly to the preference structure $(P, P_{\mathbf{C}}, P_{\mathbf{A}})$ with $P := P_2 \to P_1$, $P_{\mathbf{C}} := P_1 \vee P_2$ and $P_{\mathbf{A}} := P_1 \wedge P_2$ and its two formula forms. (As a special case, whenever $P_2 \equiv \neg P_1$, this simplifies to the structure $\overline{P} = (P_1, \top, \bot)$) $\square$

As a corollary, this tell us that we can decompose any preference structure formed via the implication form to two formulas. When visualized as truth tables (Figure 3), which we can use an alternative encoding of preference structures, these correspond to the formulas representing the ✓s and ✗s.

### 5.1 Generalized semantic loss based on preference structures

In our generalization of the semantic loss, formulas $P$ will be replaced with preference structures $\overline{P}$. For example, we can modify the logistic log form of SL in Eq 5 to be $\ell(\overline{P}, \theta, D)$ and change the semantic loss ratio $\rho_{\text{sem}}$ accordingly to operate over the formula forms of $\overline{P}$ in Eq 6. By analogy to the generalized DPA in Eq 1, we can view this logistic log form as a particular instance of a **generalized semantic loss**:

| Name | $f(\rho_{\text{sem}}, \beta) =$ | Semantic loss ratio |
|---|---|---|
| $\ell_{\text{sl-log}}$ | $-\log \sigma(\beta \rho_{\text{sem}})$ | |
| $\ell_{\text{sl-squared}}$ | $(\rho_{\text{sem}} - \frac{1}{2\beta})^2$ | $\rho_{\text{sem}} := \log \frac{\text{WMC}(\overline{P_f}; \theta)}{\text{WMC}(\overline{\neg P_f}; \theta)}$ |
| $\ell_{\text{sl-margin}}$ | $\max(0, \beta - \rho_{\text{sem}})$ | |

Table 3: Different forms of the generalized semantic loss that match the DPA losses in Table 1.

$\ell_{\text{sl}}(\overline{P}, \theta, D) := \mathbb{E}_{d \sim D}[f(\rho_{\text{sem}}(d), \beta)]$ where, like in DPA, different choices can be made about what $f$ to apply over the semantic loss ratio $\rho_{\text{sem}}$, which gives rise to several novel logics. To match the structure of DPA, we also add a weight parameter $\beta$. We define three particular versions of SL in Table 5, which we will need to apply our formal analysis to particular DPA losses in Table 1.

**How many loss functions are there?** Under this new formulation, we can view loss creation as a generative procedure, where we first select a $f$ then sample two formulas $P_1$ and $P_2$ (each denoting a unique Boolean function in $n$ variables) to create a $\overline{P}$ via Prop 1 (see also Figure 3). This view allows us to estimate the total number of definable loss functions for choice of $f$ to be doubly exponential in the number of probabilistic predictions $n$, equal to $4^{2^n}$ (i.e., the unique pairs of Boolean functions). For DPO, which involves four probabilistic predictions, this results in more than 4.2 billion variations that can be defined (how DPO is translated into a preference structure is addressed in Section 5.2).

**How is the loss space structured?** While the space of loss functions is often very large, one can structure this space using the semantics of the corresponding formulas. Below we define preference entailment and equivalence and relate these semantic notions to the behavior of the compiled losses. The following formal results (see proofs in Appendix B) give us tools for structuring the DPA loss space and informing the search for new loss functions.

We define **preference entailment** for two preference structures $\overline{P}^{(1)} \sqsubseteq \overline{P}^{(2)}$ in terms of ordinary propositional entailment ($\models$) between formula forms: $\overline{P}^{(1)} \sqsubseteq \overline{P}^{(2)} := (\overline{P_f}^{(1)} \models \overline{P_f}^{(2)} \wedge \overline{\neg P_f}^{(2)} \models \overline{\neg P_f}^{(1)})$. Below we show (proof deferred to Appendix) that losses are monotonic w.r.t. preference entailment, as in the original SL (Xu et al., 2018).

**Proposition 2** (monotonicity)**.** *If* $\overline{P}^{(1)} \sqsubseteq \overline{P}^{(2)}$ *then* $\ell_{sl}(\overline{P}^{(1)}, \theta, D) \geq \ell_{sl}(\overline{P}^{(2)}, \theta, D)$ *for any* $\theta, D$*.*

We will use later entailment to characterize the relative strength of DPA losses and visualize their relations using a representation called a **loss lattice** (see Figure 4). We also extend preference

entailment to **preference equivalence** in a natural way: $\overline{\mathsf{P}}^{(1)} \equiv \overline{\mathsf{P}}^{(2)} := (\overline{\mathsf{P}}^{(1)} \sqsubseteq \overline{\mathsf{P}}^{(2)} \land \overline{\mathsf{P}}^{(2)} \sqsubseteq \overline{\mathsf{P}}^{(1)})$, and observe that our version of semantic loss is equivalent under preference equivalence (please see Appendix B for proofs and additional formal results).

## 5.2 DECOMPILING DPA LOSSES INTO PREFERENCE STRUCTURES

The **decompilation** of a DPA loss $\ell_{\mathrm{DPA}_x}$ into a symbolic form can now be stated as finding a preference structure $\overline{\mathsf{P}}$ whose particular semantic loss $\ell_{\mathrm{sl}_x}$ is equal to $\ell_{\mathrm{DPA}_x}$, as given in Eq 7:

$$\forall D, \theta. \ \ell_{\mathrm{DPA}_x}(D, \theta) = \ell_{\mathrm{sl}_x}(\overline{\mathsf{P}}, D, \theta) \quad (7) \quad \rho_\theta = \rho_{\mathrm{sem}}, \text{ with } \frac{\rho_\theta^t}{\rho_\theta^b} = \frac{\mathrm{WMC}(\overline{\mathsf{P}_f}; \theta)}{\mathrm{WMC}(\neg \mathsf{P}_f; \theta)} \quad (8)$$

We say that a preference structure $\overline{\mathsf{P}}$ **correctly characterizes** a loss $\ell_x$ under some $\ell_{\mathrm{sl}_x}$ whenever this condition holds. Given the structure of the DPA loss (Eq 1) and the generalized semantic loss, whenever $f$ is fixed this can be reduced to finding a $\overline{\mathsf{P}}$ whose semantic loss ratio $\rho_{\mathrm{sem}}$ is equal to $\ell_x$'s core loss equation $\rho_\theta$ as shown in Eq 8.

Based on this, we define a procedure for translating the core loss equations $\rho_\theta$ in Table 2 into preference structures and $\rho_{\mathrm{sem}}$. We consider each part in turn.

**Characterizing the DPA equation class** By construction, we will assume that all the core equations for DPA losses $\rho_\theta^t$ and $\rho_\theta^b$ are expressible as certain types of **disjoint multilinear polynomials** over binary variables $\{x_i\}_{i=1}^n$, intuitively polynomials whose translation via the rules in Table A results in valid formulas

---

**Algorithm 1:** DPA to logic

**Input** : disjoint polynomial $\rho_\theta = \frac{\rho_\theta^t}{\rho_\theta^b}$

**Output:** $\overline{\mathsf{P}} = (\mathsf{P}, \mathsf{P_C}, \mathsf{P_A})$

1 $\mathsf{P}_t \leftarrow \mathrm{SEM}(\rho_\theta^t)$
2 $\mathsf{P}_b \leftarrow \mathrm{SEM}(\rho_\theta^b)$
3 $\mathsf{P} \leftarrow \mathrm{SIMPLIFY}(\mathbf{Implies}(\mathsf{P}_b, \mathsf{P}_t))$
4 $\mathsf{P_C} \leftarrow \mathrm{SIMPLIFY}(\mathbf{Or}(\mathsf{P}_t, \mathsf{P}_b))$
5 $\mathsf{P_A} \leftarrow \mathrm{SIMPLIFY}(\mathbf{And}(\mathsf{P}_t, \mathsf{P}_b))$

---

of propositional logic. Formally, such polynomials over $n$ variables are defined as any polynomial $e$ of the form $e = \sum_i e_i$ where (a) for all $i$ there exists $J_i \subseteq \{1, \ldots, n\}$ such that $e_i = \prod_{j \in J_i} \ell_{ij}$ where $\ell_{ij}$ is either $x_j$ or $(1 - x_j)$, and (b) for all $i, i'$, terms $e_i$ and $e_{i'}$ are disjoint, i.e., have no common solutions (for some $k$, one term has $x_k$ and the other has $1 - x_k$).

We note that not all preference loss functions in the preference learning literature immediately fit this format, including the original form of DPOP (Pal et al., 2024) which we discuss in Appendix D and fix through **variable copying** as shown in Table 2.

**Translation algorithm** Our translation process is shown in Algorithm 1. Given an input $\rho_\theta$, both parts of that equation are translated into logic (**lines 1-2**) via a translation function SEM. The translation is standard and its correctness can be established via induction on the rules (see the full rules in Table A): each model prediction $P_{\mathbf{M}}(\cdot)$ is mapped to a probabilistic prediction $\mathbf{M}(\cdot)$ then: $1 - \mathsf{P}$ is mapped to negation, $\mathsf{P}_1 \cdot \mathsf{P}_2$ to conjunction, and $\mathsf{P}_1 + \mathsf{P}_2$ to disjunction. **Lines 3-5** apply the implication construction from Prop 1 to create a $\overline{\mathsf{P}}$, where formulas are minimized via SIMPLIFY.

The following result establishes the correctness of our decompilation algorithm, showing specifically that our algorithm yields preference structures that satisfy Eq 8. This follows immediately from the correctness of our translation rules and the implication construction from Prop 1.

**Proposition 3** (correctness). *Given a loss equation $\rho_\theta = \rho_\theta^t / \rho_\theta^b$ where $\rho_\theta^t$, and $\rho_\theta^b$ are disjoint polynomials, Algorithm 1 returns a preference structure $\overline{\mathsf{P}}$ whose semantic loss ratio $\rho_{sem}$ equals $\rho_\theta$.*

## 6 RESULTS AND DISCUSSION

Table 4 shows the preference structures obtained from Algorithm 1 for the DPA losses in Table 2. Since the original losses were all formulated using the logistic log form of DPA, the correctness of Algorithm 1 (Prop. 3) tells us that compiling the representations in Table 4 under $\ell_{\mathrm{sl\text{-}log}}$ will yield exactly the original losses, and hence satifies Eq 7. Importantly, when the DPO symbolic form is compiled using $\ell_{\mathrm{sl\text{-}square}}$ (i.e., the squared loss form of SL), this will yield exactly IPO (Azar et al., 2023), showing how our semantic analysis is invariant to the particular choice of $f$.

## 6.1 WHAT WE LEARN ABOUT KNOWN LOSSES?

**Single model approaches have an intuitive semantics, highly constrained** Under our analysis, CPO and ORPO are both derived from the same core semantic formula P and implication first introduced in Figure 3, in spite of the superficial differences in their original form. They differ, however, in terms of the conditioning constraints $P_C$ they impose, with CPO imposing a **one-true** constraint that requires either the winner or loser to be deemed valid, whereas ORPO imposes a **one-hot** constraint where one and only one can be deemed valid. When plotted in a broader loss landscape, as shown in Figure 4, we see that both are entailed by the CEUnl baseline, yet have a non-entailing relation to one another.

| Loss | Representation $\overline{\mathsf{P}}$ |
|---|---|
| CE | $P := \mathbf{M}(x, y_w), \ P_C := \bot$ |
| CEUnl | $P := \mathbf{And}(\mathbf{M}(x,y_w), \mathbf{Not}(\mathbf{M}(x,y_l)))$ |
| CPO | $P := \mathbf{Implies}(\mathbf{M}(x,y_l), \mathbf{M}(x,y_w))$ 
 $P_C := \mathbf{Or}(\mathbf{M}(x,y_l), \ \mathbf{M}(x,y_w))$ |
| ORPO | $P := \mathbf{Implies}(\mathbf{M}(x,y_l), \mathbf{M}(x,y_w))$ 
 $P_C := \mathbf{XOR}(\mathbf{M}(x,y_l), \ \mathbf{M}(x,y_w))$ |
| DPO | $P := \mathbf{Implies}(\mathbf{And}(\mathbf{Ref}(x,y_w), \mathbf{M}(x,y_l)),$ 
 $\mathbf{And}(\mathbf{Ref}(x,y_l), \mathbf{M}(x,y_w)))$ 
 $P_C := \mathbf{Or}(\mathbf{And}(\mathbf{Ref}(x,y_w), \mathbf{M}(x,y_l)),$ 
 $\mathbf{And}(\mathbf{Ref}(x,y_l), \mathbf{M}(x,y_w)))$ |

Table 4: Formalizations of some of the losses from Table 2 shown in terms of P and $P_C$ (for succinctness, we exclude $P_A$ which can be inferred from each $P_C$ via Algorithm 1).

In general, we see that preference losses are highly constrained. This is in contrast to the losses typically used with the semantic loss, suggesting that there is much to learn by working backward from empirically successful loss functions to their semantic properties.

**There are many losses still to explore** We created new losses by modifying the conditioning constraints of existing losses. Figure 4 shows a (non-exhaustive) lattice representation of the loss landscape for single model preference approaches created by mechanically deriving new losses from the $\ell_{\text{CEUnl}}$ baseline (the most constrained) and ordering them by strict entailment (terminating in $\ell_{\text{unCPO}}$, our running example). We see different **semantic regions** emerge characterized by different formulas P, notably an unexplored region of unlikelihood losses that optimize for the negation of the loser ($\mathbf{Not}(\mathbf{M}(x, y_l))$).

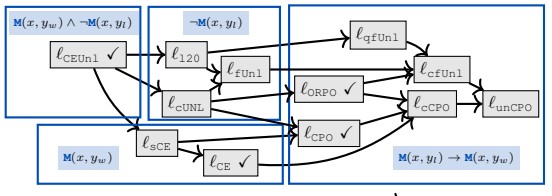

**most constrained** ⟶ **least constrained**

Figure 4: What other losses are there? Here we show the loss landscape for single model preference approaches using a **loss lattice** showing losses (nodes) structured according to strict entailment (⊑) and their core formulas P (boxes) with ✓ being the known losses. See Appendix C for details of the individual losses and Figure 5.

**DPO has a peculiar semantics, shared among variants** The semantics of DPO shown in Table 4 is logically equivalent to a conjunction of two implications: $\mathbf{Ref}(x, y_w) \land \mathbf{M}(x, y_l) \to \mathbf{M}(x, y_w)$ and $\mathbf{Ref}(x, y_w) \land \neg\mathbf{M}(x, y_l) \to \neg\mathbf{M}(x, y_l)$. The first says that *If the reference deems the winner to be valid and the tunable model deems the loser to be valid, then that model should also deem the winner to be valid*, while the second says that *the tunable model should deem the loser to be not valid whenever the reference deems the winner to be valid and the loser to be not valid*. While this semantics makes sense, and complements nicely the semantics of CPO by adding information about the referent model, DPO includes conditioning constraints that are hard to justify from first principles, and that make it semantically disconnected from the CE and CEUnl baselines.

We also note that variants like SimPO and DPOP when formalized maintain exactly the same structure of DPO in Table 4, with DPOP adding repeated variables that amplify the score of the winner. Giving the semantic similarity between these variants and DPO, any small semantic change found in one would likely be useful in these others, which motivates general exploration into varying the conditioning constraints (we show several such variants of DPO in Figure C built from Figure 4).

## 6.2 APPLYING OUR FRAMEWORK

Our formal analysis reveals that the space of DPA losses is large, yet structured in systematic ways that we can now describe through symbolic encodings. Through cases studies involving the new losses in Figure 4, we discuss some empirical results that give tips for how to better navigate this space and look for improved DPA losses using our framework. Specifically, we focus on losses around the known loss $\ell_{\text{CPO}}$, which we treat as a natural baseline to compare against. All experiments

are performed using an 0.5 billion LLM, `Qwen-0.5B` (Bai et al., 2023), tuned using `trl` (von Werra et al., 2020) on the `ultrafeedback` dataset (see full experiment details in Appendix C).

**How does constrainedness relate to loss behavior?** Moving left to the right in Figure 4 yields semantically less constrained losses. For example, we see through the Boolean semantics in Figure 5 that some unconstrained losses can be satisfied by making the winner and loser both false ($\ell_{\text{unCPO}}$, $\ell_{\text{cfUNL}}$) or by making the the winner and loser both true ($\ell_{\text{unCPO}}$, $\ell_{\text{cfUNL}}$).

We observe, consistent with other recent work on neuro-symbolic modeling (Marconato et al., 2024; van Krieken et al., 2024), that such unconstrainedness can yield extreme behavior as illustrated in Figure 5. For example, $\ell_{\text{unCPO}}$ and $\ell_{\text{cfUNL}}$ attempt to make both the winners and losers false by driving their probability in the direction of zero (as shown in in both training (a) and evaluation (b)), whereas $\ell_{\text{cfUNL}}$ keeps both probabilities high to make both true. These results suggest that understanding the way in which a loss is constrained and whether it gives rise to spurious shortcuts is an important factor when designing new loss functions.

**What is the right semantics for preference learning?** Given the spurious behavior of losses $\ell_{\text{unCPO}}$ and $\ell_{\text{cfUNL}}$, we would expect them to be less empirically successful. To test this and compare against $\ell_{\text{CPO}}$, we performed a model-as-judge-style experiment based on Hong et al. (2024) that uses an off-the-shelf reward model (Cai et al., 2024) to score the outputs generated by our new models using the prompts from the `ultrafeedback` test set. We then compare these rewards scores against those of $\ell_{\text{CPO}}$ to compute a win-rate, which gives an indication of improved generation quality over $\ell_{\text{CPO}}$. Indeed, we see in Table 5 that in aggregate, $\ell_{\text{unCPo}}$ and $\ell_{\text{cfUNL}}$ have the lowest win-rate against $\ell_{\text{CPO}}$. Interestingly, we see that $\ell_{\text{cCPO}}$ has a win-rate that suggests improved generation over $\ell_{\text{CPO}}$, which shows the potential of using our framework to derive new and empirically successful losses.

Figure 5: An illustration (A) of how to semantically satisfy losses ( ✓ ) and the corresponding log probability behavior during training (B) and evaluation (C).

| loss | WR% ($\ell_{\text{cpo}}$) | evol | false-qa | flan | sharegpt | ultrachat |
|------|------|------|------|------|------|------|
| $\ell_{\text{cfUNL}}$ | 46.1 (±0.4) | 46.1 (±2.2) | 51.6 (±2.9) | 46.4 (±1.7) | 46.2 (±1.2) | 44.1 (±1.0) |
| $\ell_{\text{qfUNL}}$ | 48.9 (±0.8) | 45.3 (±1.9) | 34.7 (±6.3) | 57.9 (±1.2) | 46.8 (±2.4) | 41.3 (±1.4) |
| $\ell_{\text{cCPO}}$ | 52.0 (±0.6) | 50.7 (±0.5) | 50.2 (±0.7) | 57.2 (±1.1) | 47.2 (±1.8) | 53.1 (±1.9) |
| $\ell_{\text{unCPO}}$ | 46.0 (±0.2) | 45.8 (±0.3) | 52.1 (±3.0) | 45.7 (±0.6) | 46.2 (±2.1) | 44.8 (±2.1) |

Table 5: Comparing performance of `Qwen-0.5B` tuned on new losses (rows) against $\ell_{\text{CPO}}$ based on aggregate win-rate (WR % (std)) on `ultrafeedback` test (second column) and different test subsets (columns 2-6).

Importantly, we see also that win-rate across different categories in `ultrafeedback` varies quite considerably across models. This suggests that different types of preference data rely on a different semantics of preference, which requires a tuning approach that's tailored to those differences. This highlights the benefit of having a framework where one can systematically study and manipulate the semantics accordingly, and we think that more empirical work in this area is a promising direction for future research.

## 7 CONCLUSION

Despite the routine use of a variety of DPA algorithms to align LLMs with human preferences, knowing what exactly the losses underlying these algorithms capture and how they relate to each other remains largely unknown. We presented a new technique for characterizing the semantics of such losses in terms of logical formulas over boolean propositions that capture model predictions. Key to our approach is the *decompilation* procedure, allowing one to derive provably correct symbolic formulas corresponding to any loss function expressed as a ratio of disjoint multilinear polynomials. Our approach provides a fresh perspective into preference losses, identifying a rich loss landscape and opening up new ways for practitioners to explore new losses by systematically varying the symbolic formulas corresponding to existing successful loss functions.

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

## A SEMANTIC TRANSLATION RULES

In Table A we show the full translation rules for Algorithm 1.

## B PROOFS OF PROPOSITIONS

| $\mathbf{M}(x, y_w)$ | $\mathbf{M}(x, y_l)$ | uncCPO | cCPO | CPO | CE | sCE | ORPO |
|:---:|:---:|:---:|:---:|:---:|:---:|:---:|:---:|
| T | T | ✓ | ✓ | ✓ ✗ | ✓ | ✓ ✗ | |
| T | F | ✓ | ✓ | ✓ | ✓ | ✓ | ✓ |
| F | T | ✗ | ✗ | ✗ | ✗ | ✗ | ✗ |
| F | F | ✓ | | | | ✗ | ✗ |

| $\mathbf{M}(x, y_w)$ | $\mathbf{M}(x, y_l)$ | cUnl | CEUnl | cfUnl | fUnl | qfUnl | l2o |
|:---:|:---:|:---:|:---:|:---:|:---:|:---:|:---:|
| T | T | ✗ | ✗ | | ✗ | | ✗ |
| T | F | ✓ | ✓ | ✓ | ✓ | ✓ | ✓ |
| F | T | ✗ | ✗ | ✗ | ✗ | ✗ | ✗ |
| F | F | | ✗ | ✓ | ✓ | ✓ ✗ | ✓ ✗ |

Figure 6: A Boolean representation (in the style of Figure 3) of the loss functions shown in Figure 4.

Below we state propositions discussed in Section 5.1 with their proofs.

**Proposition 4** (monotonicity). *If* $\overline{\mathsf{P}}^{(1)} \sqsubseteq \overline{\mathsf{P}}^{(2)}$ *then* $\ell_{sl}(\overline{\mathsf{P}}^{(1)}, \theta, D) \geq \ell_{sl}(\overline{\mathsf{P}}^{(2)}, \theta, D)$ *for any* $\theta, D$.

*Proof.* By the definition of preference entailment, we have $\overline{\mathsf{P}}_f^{(1)} \models \overline{\mathsf{P}}_f^{(2)}$. This means that for any $d$, $\overline{\mathsf{P}}^1(d) \models \overline{\mathsf{P}}^2(d)$, which implies that for any $\theta$, $\mathrm{WMC}\big(\overline{\mathsf{P}}^{(1)}(d); \theta\big) \leq \mathrm{WMC}\big(\overline{\mathsf{P}}^{(2)}(d); \theta\big)$. From the definition of preference entail-ment, we also have $\overline{\neg\mathsf{P}}^{(2)}(d) \models \overline{\neg\mathsf{P}}^{(1)}(d)$. Following a sim-

| Input | SEM$(\cdot)$ |
|:---:|:---:|
| | predictions |
| $\mathsf{P}_{\mathbf{M}}(\mathsf{y} \mid \mathsf{x})$ | $\mathsf{P} := \mathbf{M}(x, y)$ |
| | formulas P |
| $\mathsf{P}_1 \cdot \mathsf{P}_2$ | $\mathsf{P} := \mathbf{And}(\mathsf{P}_1, \mathsf{P}_2)$ |
| $1 - \mathsf{P}$ | $\mathsf{P} := \mathbf{Not}(\mathsf{P})$ |
| $\mathsf{P}_1 + \mathsf{P}_2$ | $\mathsf{P} := \mathbf{Or}(\mathsf{P}_1, \mathsf{P}_2)$ |

Table 6: Rules for the translation of loss expressions into symbolic formulas.

ilar line of reasoning as above, this implies $\mathrm{WMC}\big(\overline{\neg\mathsf{P}}^{(1)}(d); \theta\big) \geq \mathrm{WMC}\big(\overline{\neg\mathsf{P}}^{(2)}(d); \theta\big)$. Thus, for any $d$ and $\theta$, the weighted model counting ratio term in the semantic loss in Table 5 is no larger for $\overline{\mathsf{P}}^{(1)}$ than for $\overline{\mathsf{P}}^{(2)}$. It follows that $\ell_{\mathrm{sl}}(\overline{\mathsf{P}}^{(1)}, \theta, \{d\}) \geq \ell_{\mathrm{sl}}(\overline{\mathsf{P}}^{(2)}, \theta, \{d\})$. Taking the expectation over $d \sim D$, we obtain $\ell_{\mathrm{sl}}(\overline{\mathsf{P}}^{(1)}, \theta, D) \geq \ell_{\mathrm{sl}}(\overline{\mathsf{P}}^{(2)}, \theta, D)$. $\square$

**Proposition 5** (locality). *Let* $\overline{\mathsf{P}}$ *be a preference structure defined over probabilistic prediction vari-ables* $\mathbf{X}$ *with parameters* $\theta_x$. *Let* $\mathbf{Y}$ *be some disjoint set of variables with parameters* $\theta_y$. *Then* $\ell_{sl}(\overline{\mathsf{P}}, \theta_x, D) = \ell_{sl}(\overline{\mathsf{P}}, [\theta_x \, \theta_y], D)$ *for any* $D$.

*Proof.* Let $\mathbf{w}_x$ be any world over variables $\mathbf{X}$ and $\mathbf{w}_y$ be any world over (disjoint) variables $\mathbf{Y}$. Let $\mathbf{w}_{x,y}$ denote the joint world. By Eq 2, the probability of the world $\mathbf{w}_{x,y}$ in the $(\mathbf{X}, \mathbf{Y})$ space can be written as $\mathsf{P}_{\theta_x, \theta_y}(\mathbf{w}_{x,y}) = \prod_{X_i \in \mathbf{X}} Q_{\theta_x, \theta_y}(X_i) \cdot \prod_{Y_j \in \mathbf{Y}} Q_{\theta_x, \theta_y}(Y_j)$ where $Q$ is either $\mathsf{P}$ or $1 - \mathsf{P}$. Since the parameters $\theta_x$ and $\theta_y$ refer to disjoint sets of variables, we can simplify this to $\prod_{X_i \in \mathbf{X}} Q_{\theta_x}(X_i) \cdot \prod_{Y_j \in \mathbf{Y}} Q_{\theta_y}(Y_j)$.

It follows that the marginal probability of the world $\mathbf{w}_x$ in the $(\mathbf{X}, \mathbf{Y})$ space equals $\mathsf{P}_{\theta_x, \theta_y}(\mathbf{w}_x) = \sum_{\mathbf{Y}} \left( \prod_{X_i \in \mathbf{X}} Q_{\theta_x}(X_i) \cdot \prod_{Y_j \in \mathbf{Y}} Q_{\theta_y}(Y_j) \right) = \prod_{X_i \in \mathbf{X}} Q_{\theta_x}(X_i) \cdot \sum_{\mathbf{Y}} \left( \prod_{Y_j \in \mathbf{Y}} Q_{\theta_y}(Y_j) \right) = \prod_{X_i \in \mathbf{X}} Q_{\theta_x}(X_i) \cdot \prod_{Y_j \in \mathbf{Y}} \left( Q_{\theta_y}(Y_j) + (1 - Q_{\theta_y}(Y_j)) \right) = \prod_{X_i \in \mathbf{X}} Q_{\theta_x}(X_i) = \mathsf{P}_{\theta_x}(\mathbf{w}_x)$. This last expression is precisely the probability of the world $\mathbf{w}_x$ in only the $\mathbf{X}$ space. Thus, $\mathsf{P}_{\theta_x}(\mathbf{w}_x) = \mathsf{P}_{\theta_x, \theta_y}(\mathbf{w}_x)$, which implies $\mathrm{WMC}\big(\overline{\mathsf{P}}; \theta_x\big) = \mathrm{WMC}\big(\overline{\mathsf{P}}; \theta_x, \theta_y\big)$ and similarly for $\overline{\neg\mathsf{P}}$. From this, the claim follow immediately. $\square$

## C  NEW LOSSES IN LOSS LATTICE AND EXPERIMENT DETAILS

To visualize the semantics of the single model losses shown in Figure 4, we use the Boolean truth table shown in Figure 6. As illustrated Figure 3, each loss column can be mechanically converted

into a preference structure via the following steps: 1) translate $\checkmark$ and $\times$ into two standard propositional formulas that are logically consistent with the marks, $P_t$ for $P_b$, respectively, then 2) apply the rules Algorithm 1 on lines 3-5 to these formulas to get a preference structure $\overline{P}$. (Note that the formulas in boxes in Figure 4 show the core formula $P$ in the resulting preference structure and intentionally hide details about the constraints.)

With these preference structures, we can then obtain a compiled version of the loss by simply applying one of the versions of the semantic loss. In simplified terms, finding the compiled loss equation directly from a truth table for the log sigmoid SL involves the following equation:

$$-\log \sigma\left(\log \frac{\sum \boxed{\checkmark}}{\sum \boxed{\times}}\right)$$

where we can replace each $\sum .$ with the corresponding WMC equations for each mark, then simplify the resulting equation (i.e., the core loss equation) to arrive at a compact loss equation that can be directly used for implementation.

**Losses used in experiments** Employing the process above, below show the core loss equations for the losses we used in our experiments in accordance with the form in Table 2:

| **Loss name** | **Core loss equation** (implementation) |
|---|---|
| $\ell_{\text{cpo}}$ | $\log \frac{P_\theta(y_w|x)}{P_\theta(y_l|x)}$ |
| $\ell_{\text{orpo}}$ | $\log \frac{P_\theta(y_w|x)(1-p_\theta(y_w|x))}{P_\theta(y_l|x)(1-p_\theta(y_w|x))}$ |
| $\ell_{\text{cCPO}}$ | $\log \frac{P_\theta(y_w|x)}{(1-P_\theta(y_w|x))P_\theta(y_l|x)}$ |
| $\ell_{\text{qfUNL}}$ | $\log \frac{(1-P_\theta(y_l|x))}{(1-P_\theta(y_w|x)}$ |
| $\ell_{\text{qfUNL}}$ | $\log \frac{(1-P_\theta(y_l|x))}{(1-P_\theta(y_w|x)}$ |
| $\ell_{\text{cfUNL}}$ | $\log \frac{(1-P_\theta(y_w|x))}{(1-P_\theta(y_w|x))P_\theta(y_l|x)}$ |
| $\ell_{\text{unCPO}}$ | $\log \frac{p_\theta(y_l|x)p_\theta(y_w|x)+(1-p_\theta(y_l|x))}{p_\theta(y_l|x)(1-p_\theta(y_w|x))}$ |

As described above, the final loss that we implemented was then obtained by applying the logistic loss loss over these equations and adding a $\beta$ term. We used the `trl` library for implementation from von Werra et al. (2020), with assistance from the trainer scripts used in Meng et al. (2024).[4]

**Extending the loss lattice to reference models** While our loss lattice and the subsequent experiments we describe center around novel no reference loss functions, we note that given abstract structure of DPA, we can easily transform a no reference loss function into reference loss function by simply subtracting the reference log win-lose ratio, $s_{\text{ref}}(y_w, y_l)$ (either using a real reference ratio or one for simpo) from any single model loss equation (e.g., any of of the loss equations above). Via some algebraic simplification, we can then arrive a new core loss equation with this reference information and straightforwardly generate a preference structure via Algorithm 1.

Figure C shows the result of this process for the single loss functions derived in Figure 4. This reveals a wide range of novel variants of `DPO` that we leave for future experiments and study.

### C.1 EXPERIMENT SETTINGS

**Dataset and Model** Following much of the DPA work we cite, we train models on the `ultrafeedback` dataset (Cui et al., 2023), which contains around 60k binarized preference pairs aggregated from several individual preference datasets (the different categories are listed in Table 5). For tuning (detailed below) we used a custom held-out development set containing around 1.3k examples taken from the train set and reserve the test set (containing 2k examples) for final evaluation.

Standardly, we ran experiments starting from a instruction tuned model (SFT), using a `Qwen-0.5B` (containing .5 billion parameters) base model (Bai et al., 2023) that was initially tuned on 6k pairs

---

[4]see `https://github.com/huggingface/trl` and `https://github.com/princeton-nlp/SimPO`.

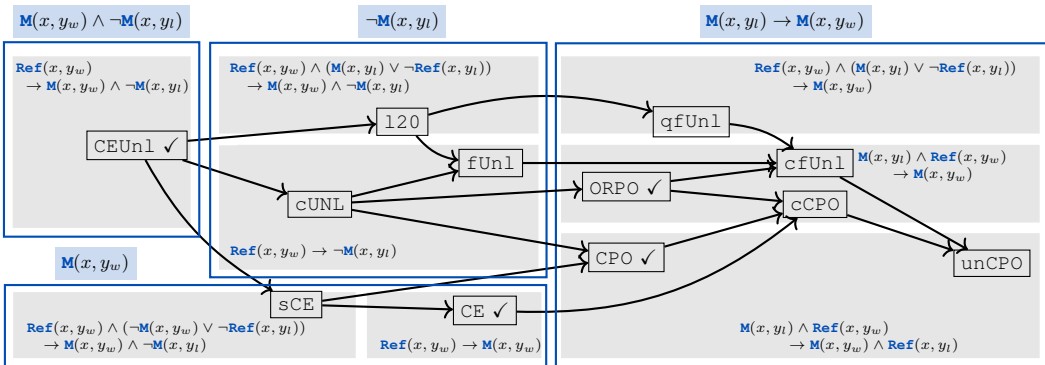

Figure 7: Extending the loss lattice in Figure 4 to a version of the single model losses with reference models, showing different (largely unexplored) variants of `DPO` and the different semantics regions (gray boxes, corresponding to the core semantic formula for P each set of losses).

from the `deita` dataset of Liu et al. (2023). To avoid repeating the process of instruction tuning, we started from the trained `Qwen` model released in the TRL library[5].

**Hyper-parameters and model selection**  The following are the standard set of tunable hyper-parameters involved in our experiments: the $\beta$ term for DPA losses (see again Table 1), the learning rate, number of epochs, batch size and length normalization. Following other studies, we also regularized our losses with cross-entropy terms (CE) that include a tunable weight parameter $\lambda$ that controls their contribution to the gradient. Specifically, we kept set $\beta$ to 1, and experimented with learning rates in the range $\{1e-6, 3e-6, 8e-6, 9e-7\}$, number of epochs in the range of $\{3, 5, 8\}$ and batches sizes in the range $\{ 32, 128 \}$ (for efficiency reasons, most tuning with done with a batch size of 32), which follow many of the suggested ranges in Meng et al. (2024). Importantly, length normalization was used throughout to make all losses comparable and given that it has been shown to improve training performance (Meng et al., 2024). We used $\lambda$s in the range of $\{0.0, 0.01, 0.1, 0.3, 1.0\}$ (we found lower values, around $0.01$ and $0.1$, to be most effective).

For each loss function we searched the best hyper-parameters by performing a comprehensive grid search over the ranges detailed above. Final model selection was then performed by performing inference with each trained model on our held-out development set and scoring the resulting generating outputs using an off-the-shelf reward model, in particular, a 1.8B parameter reward model from Cai et al. (2024)[6]. We then selected the models with the highest average reward score over the development set for comparison.

**Evaluation protocol and win-rate comparison**  We compare models tuned using our different losses using a procedure similar to how model selection is performance, which also follows the setup in Hong et al. (2024). Specifically, we do a instance-level comparison of the reward score given for each generated output, compare that score with the score of our baseline $\ell_{cpo}$ and compute an overall win-rate, i.e., % of instances where the reward score is higher than or equal to the reward score for $\ell_{cpo}$. We report the average win-rate averaged over 3 runs of each models with different generation seeds.

## D    DPOP EQUATION

The `DPOP` loss function in Table 2 adds to the `DPO` an additional log term $\alpha \cdot \max(0, \log \frac{P_{ref}(y_w|x)}{P_\theta(y_w|x)})$ that aims to ensure that the log-likelihood of preferred example is high relative to the reference model (we simplified this loss by removing the $\max$ and $\alpha$ parameter, the latter of which is set to be a whole number ranging from 5 to 500 in Pal et al. (2024)). When translating the full loss into a single log, this results in the equation $\rho_\theta = \log \frac{P_{ref}(y_l|x)P_\theta(y_w|x)^2}{P_{ref}(y_w|x)^2 P_\theta(y_l|x)}$ for $\alpha = 1$. The top and bottom

---

[5]`https://huggingface.co/trl-lib/qwen1.5-0.5b-sft`
[6]`internlm/internlm2-1_8b-reward`

equations are hence not multilinear since they both contain exponents $> 1$. To fix this, we can simply create copies of these variables, e.g., with $P_\theta(y_p \mid x)^2$ and $P_{\text{ref}}(y_l \mid x)^2$ set to $P_\theta(y_p \mid x)P_{\theta 2}(y_p \mid x)$ and $P_{\text{ref}}(y_l \mid x)P_{\text{ref2}}(y_l \mid x)$ using the copied prediction variables $P_{\theta 2}(\cdot)$ and $P_{\text{ref2}}(\cdot)$. This type of variable copying also allows us to take into account the $\alpha$ and $\max$ above by setting the values of these copied variable to be 1 whenever the log ratio is less than 0.

Below we show the core semantic formula for DPOP, which, as noted before, makes a small adjustment to the DPO semantics as shown in Table 4:

```
P := Implies(
    And(Ref(x,y),Ref₂(x,y_w),M(x,y_l)),
    And(Ref(x,y_l),M(x,y_w),  M₁(x,y_w))
)
```

