# OpenReview forum: "Declarative characterizations of direct preference alignment algorithms"
_ICLR.cc/2025/Conference — Submitted to ICLR 2025_

### Official Review · Reviewer_vR26 · 2024-10-26

**Soundness:** 3
**Presentation:** 3
**Contribution:** 2
**Rating:** 5
**Confidence:** 3

**Summary:**

The submission devises a new framework that can translate DPA loss functions into a logical characterization. As a byproduct of that framework, the authors establish a double-exponential upper-bound on the number of different DPA loss functions that can be represented in their model. The submission is fairly well-written, but it is clearly targeted at experts in the specific subfield as, e.g., the introduction assumes knowledge of DPA and DPO and is not very accessible to the general ICML audience.

I have two main concerns:

1) The contribution is comparatively weaker than what one would expect from a typical ICML paper. Of course, contributions can come in different forms, but aside from new ideas fully-fledged ICML papers typically support these ideas with a technical contribution, such as experimental evaluations or non-trivial mathematical proofs. The submission, however, lacks the former as well as the latter (all statements are established as direct observations or via essentially trivial proofs which do not seem to require novel insights or ideas).

2) While the submission identifies a large number of mathematically well-defined DPA loss functions, it is less clear what is the envisioned contribution to the ICML community. After all - unlike when one establishes, e.g., theoretical upper/lower bounds or settles the computational complexity of fundamental problems - here there was no doubt that many different DPA loss functions exist. The authors claim that their results can be used to "map out" and find loss functions with better properties, but that is left entirely for future work (see Section 6.2).

Given these concerns, I feel that the submission would benefit from diving deeper into the topic and presenting a more well-rounded and thorough contribution. Note that space constraints are not a major factor here yet: the current submission spends a lot of space discussing tangential remarks which could easily be partly or wholly moved to the Appendix (see, e.g., the end of Subsection 6.1).

**Strengths:**

The submission is well-written and the research direction seems sound.

**Weaknesses:**

See the Summary.

**Questions:**

-How important is it for the formalization to follow the assumption about DPA losses suggested by Tang et al. (arxiv 2024)? As far as I am aware, that article was not yet peer-reviewed and it is hence not clear to me how well-established this particular formalization of DPA losses is (and the submission does not attempt to justify this on its own).

---

> ### Author Response · Authors · 2024-11-23
>
> We thank the reviwer for their feedback.
>
> # two main concerns:
>
> When evaluating the contribution that our paper makes to the ICLR community, we think that it is important to place it within the context of other work on direct preference alignment, much of which is being published at venues like ICML and ICLR.  In this literature, many papers are written about a single loss function or parameterization of DPA, which has helped to empirically advance the state-of-the-art, but makes it difficult to understand the formal relationships that exist between different proposals and to discover entirely new approaches.
>
> In contrast, our approach tells us a much bigger story about the nature and structure of the target loss space. As we show as a kind of case study in Figure 5, it practically allows one to easily derive new classes of preference losses from first principles and with certain formal properties (as we mention above, we have implemented and experimented with all of these new loss functions and plan to incorporate these details in a forthcoming updated draft). As such, we think our work fills an important void in current DPA research and can help researchers in this area, who often publish at ICML and ICLR, better navigate the space.
>
> Given that our paper reveals new and interesting links between logic, weighted model counting and recent preference learning, we disagree that our paper does not make a meaningful technical contribution for this community, or that our formal results all follow from direct observation (see more discussion below).
>
> # question Tang et al and additional formal details
>
> The work by Tang et al. was published at ICML 2024, however we cite this version since it contains additional relevant details and is a longer technical report (we will cite both in the updated draft).  We note that similar formalizations have been proposed recently, including in Hu, He, Wipf et al. 2024.
>
> For our purposes, this formalization is quite helpful since it allows us to tease apart the optimization details of a loss function (e.g., the choice of convex function `f`) and the internal model quantity inside a loss (i.e., $\rho_{theta}$), the latter of which is the domain of our semantic analysis.
>
> Ultimately, this formalization allows us to prove that our semantic analysis (e.g.,  the formulas in Table 4) not only correctly characterize the target losses Table 2 (as a consequence of the correctness of translation algorithm), but does so in a way that is invariant to the choice of  `f` and different variants of DPO and CPO (e.g., those listed in Table 1). In other words, we can say that our formalization of DPO in Table 4 not only captures the semantics of the original DPO, but also the semantics of IPO by simply changing `f` and our version of semantic loss (These results are expressed verbally starting on line 456 and are probably worthy of being stated more formally as theorems, which we avoided for space and stylistic reasons).
>
> As a side note, such a generalization, which was motivated by the formal results outlined above, also give rise to the novel variants of semantic loss listed in Table 3, and hence several new logics. We believe that such logics are of independent interest to work on semantic loss and the neuro-symbolic literature more broadly.

---

> > ### Comment · Reviewer_vR26 · 2024-11-24
> >
> > I apologize for confusing ICML and ICLR in my review. Your answer has convincingly answered my question regarding the use of the formalization by Tang et al. - in particular, I see no issue with building on that formalization if it has been peer-reviewed and accepted at ICML.
> >
> > My main concern remains that, as it stands, the overall contribution seems comparatively weaker than what I would have expected from a typical ICLR paper. I will of course have a look at the revised version (including the advertised new experimental evaluations) once it is ready, and am certainly open to updating my assessment based on that.
> >
> > In line with your response, I would also encourage you to state more of the results formally (where appropriate), as proper theorems/corollaries are much easier to build on and reference in later works than long semi-formal paragraphs.

---

### Official Review · Reviewer_s3d4 · 2024-11-04

**Soundness:** 3
**Presentation:** 3
**Contribution:** 3
**Rating:** 6
**Confidence:** 4

**Summary:**

This paper addresses the challenges in understanding and developing direct preference alignment (DPA) loss functions, commonly used to align large language models with human preferences. Current DPA methods, like DPO, show promise but lack a conceptual framework for analyzing and differentiating variants. To address this, the authors propose a formalism to characterize DPA losses as discrete reasoning problems, enabling a systematic derivation of symbolic expressions that define their semantics. This approach reveals the extensive structure within the DPA loss landscape, showing a doubly exponential number of definable variations based on unique predictions. The framework highlights formal relationships, such as logical entailment and monotonicity, providing insights into efficiently exploring new DPA losses by modifying and testing established loss functions. This formal view aims to guide further development in human-AI alignment research.

**Strengths:**

The paper’s strengths include a novel technique that clarifies the semantics of DPA losses through logical formulas over Boolean propositions, addressing a key gap in understanding what these losses capture and how they interrelate. The innovative decompilation procedure enables deriving symbolic formulas for complex loss functions, offering a structured view into the DPA loss landscape. This approach empowers practitioners to systematically explore new loss functions, advancing both theoretical insights and practical tools for human-AI alignment.

**Weaknesses:**

The paper could be strengthened with more real-world examples demonstrating the practical relevance of formalizing DPA losses as discrete reasoning problems. While the formalization offers a structured approach, it’s primarily theoretical, and its effectiveness remains unproven. The claim that new losses derived from this framework are superior is speculative, as no substantial evidence is provided to show that these new losses outperform existing ones. Further empirical validation is needed to confirm the benefits and applicability of these new loss functions in real-world settings. Additionally, the hypothesis around the "constrainedness" of a loss function as a predictor of its success is only preliminary, requiring more in-depth experimentation.

**Questions:**

1. Can you provide additional real-world examples of applying formalized DPA losses as discrete reasoning problems to clarify the framework’s practical relevance?
2. While you suggest this framework aids in finding improved loss functions, is there empirical evidence or additional experiments comparing these new losses to existing DPA losses?
3. How does the complexity of your decompilation procedure scale with larger models or more complex DPA losses? This would clarify its feasibility for large-scale applications.

---

> ### Author Response · Authors · 2024-11-23
>
> We thank the reviewer for their feedback.
>
> ## practical use of framework
>
> When applied narrowly to DPA, we view our framework as a technical tool for understanding the structure of the DPA loss space, and for helping to navigate that space when looking for improved algorithms (see further comments about this below and our response to `Lvk9`). More broadly, by relying on the language of logic to express the solutions to problems (without worrying about the details of the low-level implementation of this solution and instead relying on the compilation techniques we develop), such a declarative approach  makes it easier to develop more complex algorithms. Specifically, we believe that our framework can be very helpful for expressing complex loss functions the incorporate many different components and tools, of the kind that could be relevant to tuning the kinds of multi-agent LLMs that we now build.
>
> ## empirical evidence
>
> We have implemented and run experiments involving all of the novel losses that we show in Figure 5. As mentioned at the beginning, we intend to include some of these experimental details in our updated paper draft to complement our formal results and address directly the point about  finding improved loss functions. Related to your comment above, we do see interesting connections between the constrainedness of the symbolic form of the loss and its empirical behavior, which does seem to partly explain the success of some loss functions and provides further practical advice on how to navigate the DPA space.
>
> ## complexity of decompilation
>
> For decompilation (i.e., going from a known loss function and equation to a symbolic form), the initial translation of a loss equation into a symbolic form is linear in the size of the loss equation (as implemented via the rules in Table 5). Therefore, the complexity of this part is low, especially given that most existing loss equations are compact, so this would scale linearly with more complex DPA losses (this all assumes that loss equations are expressible as the  types of multilinear polynomials that we define in line 412). The size of the model is not a factor in this context. What’s more, decompilation is an offline process that only needs to be performed once for each loss function.
>
> As we discuss in our response to `esFf` , the complexity of the `simplify` subroutine in Algorithm 1, however, does have high complexity, but is often tractable in practice, especially for the cases we consider (please see our full response for more details; we also note that this part of the Algorithm is not essential).
>
> The more complex part of our approach involves compilation (i.e., going from symbolic formulas to loss equation), which we also discuss in the response to  `esFf`.

---

> > ### Comment · Reviewer_s3d4 · 2024-11-25
> >
> > I acknowledge reading the rebuttal and thank the authors for the answers. I have no further questions at this time.

---

### Official Review · Reviewer_esFf · 2024-11-04

**Soundness:** 4
**Presentation:** 4
**Contribution:** 4
**Rating:** 8
**Confidence:** 3

**Summary:**

The paper introduces a novel framework for analyzing the space of DPA-like losses by systematically deriving a symbolic, logical expression characterizing its semantics. Different DPA losses can be derived thanks to such mapping, thus providing a comprehensive overview of the landscape. The authors do an excellent job of formalizing their framework, which gives researchers a new, fresh perspective on how to analyze the plethora of what they call successful DPA losses in the literature. Intuitively similar losses found in the literature can now have a place where analysis is done through formal methods if the paper is accepted. Be aware that my evaluation could have been overly optimistic, given my expertise; nonetheless, the paper should be accepted for me based on the novel contribution and rigorous mathematical treatment. As such, I would give a 7 (disabled by the system) instead of an 8, while a 6 seemed too pessimistic.

**Strengths:**

- **S1:** Precise related work.
- **S2:** Strong mathematical formulation, which I greatly enjoyed.
- **S3:** New perspective on neural-symbolic interplay.
- **S4:** Great presentation and structure of the work.

**Weaknesses:**

- **W1:** Given my unfamiliarity with the literature on DPA and similar, there seem to be exponential blowups. For example, to compute Eq 2, the weighted model counting must enumerate all $2^n$ propositional models ($\mathbf{w}$).
- **W2:** I would have given more context around the notation $y_w \succ y_l$ for those unfamiliar with the literature like me. I have interpreted $\succ$ as "the winner is preferred to the looser", but again, this may be the wrong interpretation. Either way, please clarify.
- **W3:** Some references are wrong. For instance, the reference to Table 7 does not exist in line 416; maybe it should be Table 5 (from the Appendix). Similarly, the caption of Table 4 references Algorithm 5.1, but 5.1 is a Section.

Minors:
- line 176: an a variant --> and a variant
- line 385 (and similar): Before the proposition's statement, the notation does not use parentheses around the superscripts, while the statement uses them (also in the Appendix). Please fix it for better readability.
- line 390: prefrence --> preference
- line 476: CEUNL --> CEUnl
- line 482: is much to learned by --> is much to *be* learned by?
- line 522: exactly these the losses --> exactly the losses?
- line 728: Table 5 is referred to in the proof, but Table 5 is the one with the translation rules
   - As a meta-observation, double-check all the other references (`\ref{}`). Some are okay, but some are not (and I tried to point to some of those).

**Questions:**

- **Q1:** I didn't get the number 4 in the double exponential form, i.e. $4^{2^n}$. Could you please be more specific or provide more context?
- **Q2:** Algorithm 1 uses "Simplify" to minimize (propositional formulas). As far as I am aware, the problem of minimizing propositional (logic) formulas by preserving equivalence is NP-hard. Thus, am I missing something here? Please clarify.

---

> ### Author Response · Authors · 2024-11-23
>
> We thank the reviewer for feedback and for the many small issues that we will fix in an updated draft. Below we address specific questions and comments.
>
> # exponential blowups, W1
>
> This is correct, the WMC semantics that the semantic loss assumes does incur an exponential blowup as you increase the number of variables. We note the following, however: the preference problems we consider have a small number of propositional variables (e.g., 2 variables for single model losses and 4 for DPO-style losses), so we do not encounter such issues in practice. Also, if we were to increase the number of variables to account for more complex losses, one can rely on well established knowledge compilation techniques that often make WMC feasible in practice (see the original paper on semantic loss for a discussion of this).
>
> # W2 and notation
>
> This is the right interpretation of this symbol $\succ$ (the winner is preferred to the loser), we will update the paper to explain this.
>
> # Q1, exponential equation
>
> Formally, the space of possible loss functions that can be expressed (or equivalently, the total number of preference structures we can define over $n$ variables) in our framework is equal to the total number of pairs of Boolean functions over $n$, which is equal to $4^{2^n}$ and where this value comes from.
>
> Intuitively, you can think of the process of coming up with a loss function generatively as sampling two arbitrary Boolean functions, one corresponding to the semantics of the “winner” and the other one corresponding to the “loser”, then compiling this into a loss via a translation into a preference structure and applying WMC to arrive at a final loss equation.
>
> To better explain this semantics visually and ground it in the set of examples and losses we show in Figures 2-3, we prepared a new figure that we will include in the forthcoming updated version of our paper.
>
> # Q2, complexity of simplify
>
> As with the complexity of general WMC, simplification is indeed a hard problem, but one that is  feasible to solve in practice for our problems (e.g., simplification of the  formulas we study can be done in milliseconds using standard computer algebra tools). More advanced SAT techniques might be used here as the problem complexity increases.
>
> We also emphasize that for each particular loss equation, Algorithm 1 is an offline process that only needs to be computed once to derive the semantics of that loss. Such complexity issues do not arise, for example, when using these losses in practice to train models. (The same is true for WMC, where the theoretically expensive step of compiling a symbolic formula into a loss via WMC only needs to be done once, since it often yields a compact formula, such as the formulas shown in Table 2, which one can implement directly and efficiently).

---

> > ### Comment · Reviewer_esFf · 2024-11-23
> >
> > I acknowledge reading the rebuttal and thank the authors for the answers. I have no further questions at this time.

---

### Official Review · Reviewer_Lvk9 · 2024-11-12

**Soundness:** 3
**Presentation:** 2
**Contribution:** 2
**Rating:** 5
**Confidence:** 3

**Summary:**

The authors investigate losses for preference alignment. They analyze existing DPO functions with an aim to symbolically extract their semantics, and also investigate compiling DPO losses from symbolically given knowledge on preference structures.

**Strengths:**

- The authors systematically investigate an interesting and relevant problem of semantically understanding and constructing preference alignment loss functions.
- The paper proposes a simple  algorithm to compile DPA loss to a logical expression
- They introduce a logic for modeling preferences, that allows to create new loss functions for a given preference structure.

**Weaknesses:**

- This could be due to my relative lack of expertise in the field of the paper. But the lack of any running example, and experiments, makes the presentation quite divorced from the original motivation of preference alignment in AI models.
- Hence, the larger potential utility of the framework is not clear to me.

**Questions:**

- Could you please provide a toy example, and an analysis of this example for each of the introduced contribution of the paper?
- What could be an empirical setting, where your proposed framework could be investigated?
- Could you please elaborate on "While this can be remedied by modifying the SL to involve counting multiple formulas as in Rescher (1967), we instead define a relational structure called a preference structure that allows us to capture the semantics of losses in a modular fashion using a single propositional formula coupled with auxiliary constraints. Such a structure, which is based on a novel construction in propositional logic, will later make it easy to cleanly characterize different DPA losses and devise new variants through manipulation to their constraints." --- It is not clear to me, why the new method you propose is motivated by this. Aren't you compiling your loss to an SL as well? what are the main differences?

---

> ### Author Response · Authors · 2024-11-23
>
> Thanks for the feedback. Below we are address your points in turn.
>
> # weakness, running example, question 1
>
> We have plans to add a new figure, specifically one that better illustrates the semantics of WMC and preference structures and connects it with the running examples shown in Figure 2 and Figure 3 (this figure will appear in our updated draft).
>
> We address the topic of experiments below.
>
> # empirical setting for testing framework
>
> Our broader goals do not deviate from the goals of other work on direct preference alignment (DPA), which aim to find novel loss functions that empirically advance the current state-of-the-art. Our view, however, is that achieving this goal requires a semantic framework that allows one to derive loss functions from first principles and to better understand the structure of the target loss space; such a framework, in our view, is missing from current work, including in the theoretical work we cite.
>
> We believe that our proposed framework achieves this first technical goal. As we show in Figure 3, it allows us to now derive entirely new families of loss functions that one can experiment with and that, we argue, would be difficult to derive without the semantic machinery we introduce. As part of our study, we also implemented all the novel loss functions shown here (i.e., nodes in Fig 5 without checkmarks, which are further defined in the appendix) and plan to include some of these auxiliary experimental results in the updated draft of our paper.
>
> # question about semantic loss
>
> As we mentioned in this paragraph you cite (starting line 300), the standard semantic loss (SL) assumes that loss functions are expressible as a single propositional formula (or equivalently, as the log ratio of model counts of a formula and that formula’s negation as per the derivation in Eq 4.).
>
> One of our early technical observations is that none of the standard preference loss functions in Table 2 (excluding the baselines) can be expressed in these terms, hence the standard SL cannot be used to do our target analysis. A somewhat technical explanation of this with an example is given in Footnote 2.
>
> Intuitively, it relates to the fact that the semantic formulas that express information about winners and losers in existing losses when translated to logic are often not logically connected to one another, thus making it not always possible to express one as the negation of the other and requires multiple semantic formuals (e.g., the logical propositions that underlie the CPO loss: “the winner is valid generation for x” and “the loser is a valid generation” express two separate facts that are not negations of the other). This issue is discussed in the logical work that we cite and we hope that the figure we mentioned above will help clarify some of the confusion here.
>
> In general, our generalized form of SL is therefore motivated by these facts and extends the standard SL in interesting ways. The preference structure we define is a convenient  (and, as we prove, a mathematically correct) way to represent multiple formulas in a way that allows us to discern general relationships between the semantics of the losses we study (e.g., in terms of these semantic neighborhoods, or boxes, that we illustrate in Figure 3).

---

> ### Comment · Reviewer_Lvk9 · 2024-11-24
> **I am still not convinced, especially about SL not being able to express DPA's**
>
> - You mention: "As it turns out, none of the variations of DPO and their log ratios in Table 2 can be expressed as a single formula in standard SL.4 While this can be remedied by modifying the SL to involve counting multiple formulas as in Rescher (1967), we instead define a relational structure called a preference structure that allows us to capture the semantics of losses in a modular fashion using a single propositional formula coupled with auxiliary constraints. Such a structure, which is based on a novel construction in propositional logic, will later make it easy to cleanly characterize different DPA losses and devise new variants through manipulation to their constraints." --- The first and second sentence are contradictory to each other. If your point is that you need to add auxiliary constraints, then yes I agree that must be done. But this is not a deep observation. Almost all of logic programming languages, essentially add a new auxiliary variable once you add a new rule --- I think this claim should be relaxed to "we provide a new encoding". Infact this is I reckon is what you do in equation 6. In order to make a statement about the fact that DPA can not be expressed in semantic loss, you would need to show that for no amount of new symbols and real parameters, one can express a function in semantic loss, and I do not see any such proof in your paper.
>
> - I will be willing to reconsider my score if you add more examples and experiments, but at the moment, I feel the general confusion due to lack of any motivating example is shared by other reviewers as well.

---

> ### Author Response · Authors · 2024-11-25
>
> `In order to make a statement about the fact that DPA can not be expressed in semantic loss, you would need to show that for no amount of new symbols and real parameters, one can express a function in semantic loss, and I do not see any such proof in your paper.`
>
> Thanks for pointing this out! We agree that one has to be careful with the language here and that **our argument is weaker and rests on certain assumptions about how formulas are constructed when using the semantic loss** (SL) in our framework (assumptions that, we believe, will be intuitive to practitioners in this area).  We will soften the language accordingly.
>
> For the sake of clarity, below is a more formal version of the argument.
>
> First, given any preference loss function $\ell$ (e.g., any of the losses in Table 2) that we want to **decompile** into a symbolic form $\mathsf{P}$ **using the SL** (via Eq. 5), we assume that:
>
> 1. All model predictions in that loss (i.e., explicit forward model calls) denote **atomic propositions**;
> 2.  (as stipulated in our description of propositional formulas starting on *line 232*) The propositional formulas $\mathsf{P}$ used to express $\ell$ in SL are limited to formulas **defined solely over those explicit atomic propositions** and none others (hence, no auxiliary variables are allowed or assumed).
>
> This second assumption is limiting, but this seems like a reasonable restriction to start with since it is not *a priori* obvious when looking at a loss what those additional variables should be (see more discussion below).
>
> Our claim is then that the losses in Table 1 cannot be expressed in SL in the sense that in each case **there does not exist a single propositional formula with the properties above that can be compiled back into that loss via SL** (the part about requiring a single propositional formula is simply what the original SL demands by definition. If we define a version involving 2 arbitrary formulas, then this is no longer the standard SL).
>
> **proof sketch**  (*a version of the argument in Footnote 1*) We can show this using an example loss, $\ell_{\text{CPO}}$, which is defined as $-\log \sigma( \log \frac{w}{l})$, where $w$ and $l$ intuitively correspond to the predictions for *winner* and *loser* and will also be used to denote our atomic propositions. Given the restriction above, we are limited to searching all propositional formulas $\mathsf{P}$ that are defined solely over the atoms $w$ and $l$.
>
> Our claim is that **no such formula exists** s.t., $\ell_{\text{CPO}} = p_{\theta}(\mathsf{P})$ (it is useful here to use the logistic log form of SL we derive in Eq. 5). This can be seen by enumerating all possible 16 Boolean functions over $w$ and $l$ and checking that none yield a propositional formula that satisfies this equivalence. The same can be done for all the other losses in Table 2.
>
> **You are right** that we have not proven that *no propositional* formulas exist that would allow one to express each DPA loss in SL (e.g., ones that introduce additional variables or involve additional transformations). We note, however, that we did make an earnest attempt to derive these losses via additional constraints to no avail and the known transformations, including the ones you mentioned in logic progrmaming, didn't seem helpful here. So we think that the solution to this, if it exists, is not obvious, which motivated us to come up with preference structures, which are, **as you suggest**, a different way of encoding the problem that gives rise to a novel form of SL.

---

> ### Author Response · Authors · 2024-12-02
> **Continued**
>
> > why … [do preference structures] create better intuition about preference losses?
>
> **example** Let’s take a particular example, the semantics of the losses $\ell_{\text{CPO}}$ and $\ell_{\text{ORPO}}$ as shown in `Figure 3`. We can express each loss as being proportional to the log ratio of the weighted model counts of two propositional formulas, i.e., any two formulas representing the checkmarks and the x marks in each column. Based on these 4 formulas, however, it is not easy to discern how these two losses are semantically related to one another, since in this case they don’t have a clear entailment relationship.
>
> Preference structures aim to bring out certain symmetries between these kinds of losses. In a preference structure, each loss is expressed as a core propositional formula $\textsf{P}$ coupled with a set of auxiliary constraints. For example, it allows us to express  $\ell_{\text{CPO}}$ and $\ell_{\text{ORPO}}$ as having the same core formula $$\textsf{l} \to \mathsf{w}$$
> but being different in terms of the constraints $\textsf{P}_{\text{C}}$ they impose, which place limits on the kinds of propositional models that can be counted. In this case,  ORPO  imposes a one-hot constraint $\textsf{w} \oplus \textsf{l}$ (which excludes counting models where both the winner and lower are true or false) and CPO imposes a weaker constraint $\textsf{l} \lor \textsf{w}$ (which excludes counting models where the winner and loser are both false).
>
> This structure or encoding is convenient for not only revealing these symmetries, but also for deriving new losses. To come up with a new loss, one can simply modify the auxiliary constraints, e.g., by removing constraints altogether, this yields a hitherto unknown loss $\ell_{\text{unCPO}}$ that can in principle be used for experimentation (as we do in `Sec 6.2`). Importantly, the proof of `Prop 1` gives us a particular encoding, the *implication constuction* that can be used to compile any two propositional formulas into such a preference structure.
>
> > why are these good semantics?
>
> Becuase they bring out the kinds of relationships we discuss above, and (we believe) help to derive new losses in an intuitive way. Importantly, we think that they are good because they are correct, i.e., can be compiled exactly into the target loss functions we care about.
>
> > most natural structure to look into … [involves] partial orders
>
> Here’s another view of the problem, which is familiar from the literature on preference logics we cite (e.g., `Rescher 1967`, stemming from the seminal work of `von Wright 1963`). Under the logical formulation above, we can say that our goal is to model a propositional preference relation between two propositions $\textsf{w} P^{\mu} \textsf{l}$, which holds when the score of $\textsf{w}$ exceeds that of $\textsf{l}$, or $\mu(\textsf{w}) > \mu(\textsf{l})$ under some scoring function $\mu$. $P^{\mu}$ is usually assumed to be a (strict) partial order, which is a property that sometimes follows straightforwardly from the choice of $\mu$ (e.g., if WMC is used for $\mu$, as we do,  such properties will be satisfied as discussed further in `Rescher 1967`).
>
> There is nothing incompatible with these approaches and our approach, especially given that they are couched in the same possible world semantics that we use. We could more explicitly base our formalization around such a relation, and perhaps this is worthwhile to do in future work, but it’s not clear how this helps us to solve the problems we described above and how this is a natural construction for our purposes.
>
> **[von Wright, Georg Henrik: 1963, The Logic of Preference, Edinburgh University Press, Edinburgh]**
>
> ## past questions
>
> > the things that I did manage to understand, were in some way either flawed or not a very deep observations
>
> Can you clarify which parts seem flawed and what you mean by *not very deep observations*?
>
> > semantic loss is not equivalent to WMC
>
> We do acknowledge this fact and note that our particular variant of WMC is clearly defined in `Eq. 2` with probabilities. Do you think it’s misleading to refer to this notionally as `WMC`? (We’d be happen to change this if so, since we also see the potential for confusion here. Short of this, we did change the text in line 238 to specify that we use a *variant* of weighted model counting to avoid confusion).
>
> > I am not sure if anything that you derive can not be done with existing encodings
>
> Can you give some technical intuitions for why you think existing encodings can be used in our case? Being very familiar with the work you cite, it’s really not clear how they kinds of encodings are applicable.
>
> Also, if existing encodings with additional variables were to be applicable, why would this be an improvement over the solutions we have (i.e., truth table representations or preference structures), which don't involve extra variables?

---

> ### Comment · Reviewer_Lvk9 · 2024-12-02
> **A naive encoding which is arbitrarily expressive**
>
> Dear Authors,
>
> I really appreciate that you are engaging with me and answering my (at times) repetitive questions. I think a lot of what we are discussing is subjective. However, I guess we can objectively discuss merits of the introduced propositional encodings. Here is my attempt.
>
> I may have failed to understand all the details of the paper, but I guess you want to encode constraints in WMC/SL. Your argument is that they are not expressible in WMC/SL, without additional variables. I can not argue more for merits of admitting additional variables than saying that they are ubiquitous in many works that aim to merge logic and probability --- some of them you cite [1] and others can practically automate the procedure of adding auxiliary variables, not much complex expert knowledge is needed for this.
>
> **Arbitrarily Expressive Semantic Loss encoded as WMC**
>
> What is Arbitrarily Expressive SL?
>
> Given a set of worlds $\Omega$, I define an arbitrarily expressive SL as follows:
>
> $$SL(P) = \sum_{\omega \models P} p(\omega)$$
>
> SL is arbitrarily expressive  if I can write any possible value $p(\omega)$ for each $\omega$.
>
> *I will now encode such an SL in WMC*
>
> I am giving you a worst-case encoding, that encodes arbitrarily expressive SL into WMC.
>
> Let us assume that you have a set of possible worlds $\Omega$ in your propositional language. Your goal is to assign an arbitrary weight to each of these models. Each of the models can be represented as conjuncts of all literals $l$ such that $\omega \models l$, and in your case I reckon number of propositional variables is not really a concern.  Now, for each $\omega \in \Omega$, you can introduce a new propositional variable $c_{\omega}$, and you encode $\omega$ into a conjunct $C_{\omega}$, each literal in the language of $\Omega$ gets a weight 1 in the WMC. Now,   $c_{\omega}$ gets a weight $p_{\omega}$, and negation of $c_{\omega}$ gets weight $1$ --- note that I am in WMC, so only requirement is real valued weights not probabilities. You can easily simulate any constraint $P$, by setting $c_{\omega} = 0$ if $\omega \not\models P$.
>
> **CLAIM: The following WMC formula encodes an arbitrarily expressive SL on the possible worlds in $\Omega$.**
>
> $$\mathrm{WMC}(\land_{\omega} \big( c_{\omega} \leftrightarrow C_{\omega} \big), \(p_{\omega}\)_{\omega}) $$
>
> **Proof:**
> Let us define $\Omega'$ to be the set of extended worlds with $c_{\omega}. $Let us define $\land_{\omega} \big(c_{\omega} \leftrightarrow C_{\omega}\big) $ to be $\Phi$. Set weights of all literals in the entire language to 1, except,  if you want to assign probability $p_\omega$ to the world $\omega$ (in the original language), then just set $c_{\omega}$'s weight to $p_{\omega}$. Note that this $p_{\omega}$ can be parameterized by the prediction values of the NN as well.
>
>
> Observation1: Each $\omega \in \Omega$ extends to a unique model of $\Phi$, i.e., the model where $c_{\omega}$ is true. Hence, each model in the intended SL is counted only once in the WMC.
>
> Observation 2: Any model counted in WMC is a unique extension of a model in $\Omega$. This is because  a model counted in WMC, will satisfy at least one $C_{\omega}$ --- because all $C_{\omega}$ false is a contradiction, and hence has to have atleast one $c_{\omega}$ to be true, due to how $\Phi$ is defined.
>
> Observation 3: If an extension of $\omega$ is counted in WMC, then its weight is $p_{\omega}$. Any such extension is a model of  $\Phi$ iff $c_{\omega}$ is true and $C_{\omega}$ is satisfied. They constitute a complete assignment,  and  contribute a weight $p_{\omega}$.
>
>
> I may have missed something in writing this encoding, but my point is that such encodings are routine. If this reduction is not interesting then one may look at [2]. Note that [2] discusses MLNs, but all MLNs can also be expressed as WMC.
>
> Please let me know what aspects of this does your encoding improve? or what does these encodings not capture?
>
> This summarizes my intuition for why the problem you address is already solved by existing encodings. You can automatically check WMC and entailment with this encoding, with conventional solvers. About, clearer semantics, I think this is where subjectivity comes into play. I still do not see why the encoding you introduce is more useful than the one here. Note that this is a worst-case encoding, you can make it more succinct with lesser constraints.
>
> [1] On probabilistic inference by weighted model counting. Chavira and Darwiche
>
> [2] Markov Logic Networks. https://homes.cs.washington.edu/~pedrod/papers/mlj05.pdf

---

> > ### Author Response · Authors · 2024-12-03
> >
> > > Arbitrarily Expressive Semantic Loss
> >
> > Thank you for these details, they really help to better understand your points.
> >
> > Yes, we do see how such an encoding, which is familiar to us, might be used here. It's still unclear to us how introducing $2^n$ new variables (i.e., $c_{\omega}$), as your encoding does, is preferable to one that doesn't. Moreover, the semantics of the encodings you describe seem to largely reside in the (exponentially many) real-valued *weights* (one for each $\omega$) of the formula as opposed to the *logic* in the underlying Boolean formula, making it unclear how this formulation is any more useful than the original loss function itself.
> >
> > The Boolean formulas underlying our encoding are, importantly, *unweighted* and thus more readily interpretable. For instance, the (unweighted) formula capturing CPO is simply $Implies(loser, winner)$ under the conditioning constraint that at least one of loser and winner is predicted to be true; there are no weights. This makes it possible to draw certain semantic relationships that help our particular use cases (e.g., reasoning about logical entailment between losses, deriving new losses).
> >
> > > I am in WMC, so only requirement is real valued weights not probabilities
> >
> > It is worth pointing out that this general form of WMC is at odds with the variant of WMC used in the original semantic loss and in standard probabilistic logic, where weights are instead assigned in a way that defines a probability distribution over all worlds. While one might have other motivations for doing this and it is perhaps worth exploring, it does change significantly the meaning of the counts you get (e.g., they no longer correspond to formula probabilities).
> >
> > > Your argument is that they are not expressible in WMC/SL, without additional variables
> >
> > Our argument is a little more subtle, here it is again with an example.
> >
> > Suppose we have the loss $\ell_{\text{CPO}}$ from before, defined as follows:
> > $$\ell_{\text{CPO}} = -\log \sigma \bigg( \log \frac{ p_{\theta}(\textsf{w}) }{ p_{\theta}(\textsf{l}) } \bigg)$$
> > where we (again) use $p_{\theta}(\textsf{w})$ and $p_{\theta}(\textsf{l})$ to denote the winner and loser predictions and their probabilities, respectively. To translate this into semantic loss, our goal (as a reminder) is to find a single propositional formula $\textsf{P}$ s.t. the following equalities hold:
> > $$\ell_{\text{CPO}} = -\log \sigma \bigg(\log \frac{WMC_{\theta}(\textsf{P})}{WMC_{\theta}(\neg \textsf{P})} \bigg) = -\log \frac{WMC_{\theta}(\textsf{P})}{WMC_{\theta}(\textsf{P}) + WMC_{\theta}(\neg\textsf{P})} = \underbracket{-\log WMC_{\theta}(\textsf{P})}_{\text{standard version of SL}}$$
> >
> > where, importantly, the last equality only holds when we employ the particular variant of $WMC_{\theta}$ that involves weights that define a probability distribution over all worlds (this is due to the denominator summing to 1 in the third equation).
> >
> > (**our claim**) Our initial claim is that such a $\mathsf{P}$ cannot exist that satisfies these equalities, hence making $\ell_{\text{CPO}}$ not expressible via standard SL. We concede again that our initial phrasing of this claim was problematic without clearly stating the assumptions about WMC and how formulas can be built
> > (*we did change this in the updated draft and plan to be even more formal and specific in the next version, with mention of the possibility of using the kinds of encodings you suggest*).
> >
> > **Your suggestion** does make it seem possible to arrive at a single propositional formula that satisfies the first equality under general WMC. For example, we could use a single variable, let's call it $A$, and assign it a semantics where $A$ being true corresponds to *the model deems the winner to be a good prediction* (i.e., the proposition $\textsf{w}$ we used before) and $A$ being false, or $\neg A$, corresponds to *the model deems the loser to be a good prediction* (i.e., $\textsf{l}$ from before).  With this single variable we would then have two possible worlds, one where $A$ is true *arbitrarily* weighted by $p_{\theta}(\textsf{w})$ and the other where $\neg A$ is true weighted by $p_{\theta}(\textsf{l})$. Adding your auxiliary variables $c_{i}$ and weighting variables in the manner you suggested, counting $A$ would seem to satisfy the first two equalities (in the equation above) but not quite the last one. In any case, as we discuss above in this response, the "semantics" in this WMC formulation would lie mainly in the real-valued weights of the weighted formula rather than in its logic, which we think makes the resulting formula less interpretable.

---

### Author Response · Authors · 2024-11-23

We thank the reviewers for their insightful comments and feedback. Below we address individual questions in textual form and will release an updated version of our paper in the coming days (we specifically plan to include some experimental results that connect parts of our formal framework with the empirical  behavior of the new set of loss functions that we derive in Figure 3)

---

> ### Author Response · Authors · 2024-11-28
> **updated draft**
>
> We thank again all the reviewers for their feedback.
>
> **We just updated our draft to account for the different points that came up during the rebuttal.** Below are details about the major changes (which are marked in blue in the PDF).
>
> - We included explicit running examples in the different figures (e.g., `Fig2`, `Fig3`, `Fig4`, `Fig5`) and added text to make them more coherently fit together.
>
> - We introduced a new figure, `Figure 3`, that attempts to better illustrate our semantic loss in terms of  Boolean truth tables. More details are included in the appendix about how to translate between such tables and preference structures (`Appendix C`).
>
> - (**most substantially**) We introduced a new section, `6.2`  that includes a case study and some experiments related to the new losses we show in `Figure 4`. The goal of this section is to address directly questions about the applications of our framework and show its potential to help find improved DPA losses.
>
> Through fairly standard  preference tuning experiments (details are in `Appendix C.1`) we highlight some interesting relationships we see between our formal analysis and the training behavior of some of the losses next to $\ell_{\texttt{CPO}}$ in `Figure 4` .  We also compare the generation performance of these new losses against $\ell_{\texttt{CPO}}$ using a model-as-judge style evaluation and found one of our new losses ($\ell_{\texttt{cCPO}}$) to have competitive performance (`Table 5`).

---

### Meta-Review · Area_Chair_zbqp · 2024-12-17

**Metareview:**

Reviewers believed that the topic is important and appreciated the novel neural-symbolic approach. On the negative side, some reviewers are concerned with the relevance and significance of the contributions. There were a lot of discussions between the authors and reviewers during the rebuttal phase, which clarified many points. This greatly helped the (slightly) negative reviewers to have a deeper understanding of the contributions, yet they still believe that the contributions are not significant enough to clear the bar.

**Additional Comments On Reviewer Discussion:**

Reviewer Lvk9 should be nominated for a reviewer award. He/she extensively engaged in the discussions and provided convincing reasonings behind his/her recommendations. The most positive reviewer esFf did not object to rejection.

---

### Decision · Program_Chairs · 2025-01-22

Reject